# Version 4 CALIPSO IIR ice and liquid water cloud microphysical properties, Part I: the retrieval algorithms

Anne Garnier[1], Jacques Pelon[2], Nicolas Pascal[3], Mark A. Vaughan[4], Philippe Dubuisson[5], Ping Yang[6], and David L. Mitchell[7]

[1]Science Systems and Applications, Inc., Hampton, VA 23666, USA
[2]Laboratoire Atmosphères, Milieux, Observations Spatiales, Sorbonne University, Paris, 75252, France
[3]AERIS/ICARE Data and Services Center, Villeneuve d'Ascq, 59650, France
[4]NASA Langley Research Center, Hampton, VA 23681, USA
[5]Laboratoire d'Optique Atmosphérique, Université de Lille, Villeneuve d'Ascq, 59655, France
[6]Department of Atmospheric Sciences, Texas A&M University, College Station, TX 77843, USA
[7]Desert Research Institute, Reno, NV 89512-1095, USA

*Correspondence to*: Anne Garnier (anne.emilie.garnier@nasa.gov)

## Abstract

Following the release of the Version 4 Cloud-Aerosol Lidar with Orthogonal Polarization (CALIOP) data products from the Cloud-Aerosol Lidar and Infrared Pathfinder Satellite Observations (CALIPSO) mission, a new version 4 (V4) of the CALIPSO Imaging Infrared Radiometer (IIR) Level 2 data products has been developed. The IIR Level 2 data products include cloud effective emissivities and cloud microphysical properties such as effective diameter and ice or liquid water path estimates. Dedicated retrievals for water clouds were added in V4, taking advantage of the high sensitivity of the IIR retrieval technique to small particle sizes. This paper (Part I) describes the improvements in the V4 algorithms compared to those used in the version 3 (V3) release, while results will be presented in a companion (Part II) paper. The IIR Level 2 algorithm has been modified in the V4 data release to improve the accuracy of the retrievals in clouds of very small (close to 0) and very large (close to 1) effective emissivities. To reduce biases at very small emissivities that were made evident in V3, the radiative transfer model used to compute clear sky brightness temperatures over oceans has been updated and tuned for the simulations using MERRA-2 data to match IIR observations in clear sky conditions. Furthermore, the clear-sky mask has been refined compared to V3 by taking advantage of additional information now available in the V4 CALIOP 5-km layer products used as an input to the IIR algorithm. After sea surface emissivity adjustments, observed and computed brightness temperatures differ by less than ± 0.2 K at night for the three IIR channels centered at 08.65, 10.6, and 12.05 µm, and inter-channel biases are reduced from several tens of Kelvin in V3 to less than 0.1 K in V4. We have also improved retrievals in ice clouds having large emissivity by refining the determination of the radiative temperature needed for emissivity computation. The initial V3 estimate, namely the cloud centroid temperature derived from CALIOP, is corrected using a parameterized function of temperature difference between cloud base and top altitudes, cloud absorption optical depth, and CALIOP multiple scattering correction factor. As shown in Part II, this improvement reduces the low biases at large optical depths that were seen in V3 and increases the number of retrievals. As in V3, the IIR microphysical retrievals use the concept of microphysical indices applied to the pairs of IIR channels at 12.05 µm and 10.6 µm and at 12.05 µm and 08.65 µm. The V4 algorithm uses ice look-up tables (LUTs) built using two ice habit models from the recent "TAMUice 2016" database, namely the single hexagonal column model and the 8-element column aggregate model, from which bulk properties are synthesized using a gamma size distribution. Four sets of effective diameters derived from a second approach are also reported in V4. Here, the LUTs are analytical functions relating microphysical index applied to IIR channels 12.05 µm and 10.6 µm and effective diameter as derived from in situ measurements at tropical and mid-latitudes during the TC4 and SPARTICUS field experiments.

# 1 Introduction

An accurate retrieval of cloud microphysical properties at the global scale is important to present day questions on Earth radiation and cloud forcing in climate change (e.g.; Bodas-Salcedo et al., 2016; Muhlbauer et al., 2014). The A-Train international constellation of satellites (Stephens et al., 2002) has delivered a broad range of new insights by gathering observations from multiple sensors operating in the visible/near infrared (0.4 – 8 µm) and infrared (8-15 µm), and by offering complementary measurements acquired simultaneously by both active and passive sensors (Stephens et al., 2018; Duncan and Eriksson, 2018; Stubenrauch et al., 2021). The combination of passive infrared and active instruments enables the daytime and nighttime retrievals necessary to investigate diurnal changes. The quality of A-Train data records is continuously improving due to the mutual benefit of simultaneous observations. Observations of cloud properties in the thermal infrared are available from MODIS (Heidinger et al., 2015) as well as from the hyperspectral Atmospheric Infrared Sounder (AIRS) (Kahn et al., 2018), further allowing profiling capabilities from multiple spectral channel analysis. Since the co-manifested launch of the Cloud-Aerosol Lidar and Infrared Pathfinder Satellite Observations (CALIPSO, Winker et al, 2010) and CloudSat (Stephens et al., 2018) in 2006, combined lidar-radar observations have been used for the retrieval of microphysical ice cloud properties (DARDAR, see Delanoë and Hogan (2008, 2010) and 2C-ICE, see Deng et al. (2010)). The CALIPSO Cloud-Aerosol Lidar with Orthogonal Polarization (CALIOP) and Infrared Imaging Radiometer (IIR) have provided new insights into ice cloud properties (Garnier et al., 2012, 2013, hereafter G12 and G13). Using an improved split-window technique based on its three medium-resolution channels at 08.65 µm, 10.6 µm, and 12.05 µm, IIR provides three main properties of clouds, namely effective emissivity, effective diameter ($D_e$) and ice water path (IWP). IIR is co-aligned with CALIOP in a staring near-nadir looking configuration. The center of the 69-km IIR swath is by design co-located with the CALIOP ground track so that each IIR 1-km track pixel includes three successive 100-m CALIOP footprints separated by about 333 m. Since the beginning of the CALIPSO mission, combined IIR and CALIOP observations have been used to derive multi-sensor data products that take full advantage of the quasi-perfectly co-located measurements, using the high detection sensitivity and accurate geometric altitude determination provided by CALIOP to inform the IIR radiance inversion analysis for both day and night.

Effective emissivities and microphysical retrievals are reported in the IIR Level 2 data products. The Version 3 (V3) products released in 2011 used the V3 CALIOP data products. As described in G12 and G13, they were focused on retrievals of ice cloud properties. Effective emissivity in each IIR channel represents the fraction of the upward radiation absorbed and re-emitted by the cloud system. The IIR 1-km pixel is assumed to be fully cloudy and the qualifying adjective "effective" refers here to the contribution from scattering. The retrievals are applied to suitable scenes that are identified and characterized by taking advantage of co-located CALIOP retrievals. Effective emissivity is retrieved after determining the background radiance that would be observed in the absence of the studied cloud system, and the blackbody radiance that would be observed if the cloud system were a blackbody source. Unlike the well-known split-window technique (Inoue, 1985), which relies on the analysis of inter-channel brightness temperature differences, IIR microphysical retrievals use the concept of microphysical index ($\beta_{eff}$) proposed by Parol et al. (1991). This concept is applied to the pairs of IIR channels at 12.05 µm and 10.6 µm and at 12.05 µm and 08.65 µm, with $\beta_{eff}12/10$ and $\beta_{eff}12/08$ defined as, respectively, the 12.05-to-10.6 ratio and the 12.05-to-08.65 ratio of the effective absorption optical depths. The latter are derived from the cloud effective emissivities retrieved in each of the three channels. The microphysical indices are interpreted in terms of $D_e$ by using look-up tables (LUTs) built for several ice habit models. $D_e$ is retrieved using the ice habit model that provides the best agreement with the observations in terms of relationship between $\beta_{eff}12/10$ and $\beta_{eff}12/08$. Total water path is then estimated using IIR $D_e$ and visible optical depth estimated from IIR effective emissivities. Retrievals along

the CALIOP track are extended to the IIR swath by assigning to each swath pixel the retrievals in the radiatively most similar track pixel at a maximum distance of 50 km (G12). This most similar track pixel is found by minimizing the mean absolute difference between the brightness temperatures in the three channels, with an upper threshold set to 1 K. Retrievals along the CALIOP track and over the IIR swath are reported in the IIR Level 2 track and swath data products, respectively. Accurate retrieval of emissivities from infrared radiometric inversion has proved to be valuable in providing useful complementary retrievals for inferring possible biases in methodological approaches (Garnier et al., 2015, -hereafter G15-, Holz et al., 2016), and for retrieving optical depths and microphysical properties (G13, Mitchell et al., 2018-hereafter M18). It was further shown in M18 that realistic satellite retrievals of ice concentration, $N_i$, would provide a powerful constraint for parameterizing ice nucleation in climate models. The retrieval of $N_i$ as a function of geographic area is of particular importance as it provides insight into specific interaction processes controlling cloud concentration, showing the importance of homogeneous ice nucleation under relatively clean (i.e. relatively low aerosol optical depth) conditions (M18), or the formation of liquid clouds from activated aerosol particles and indirect effect analysis (Twomey, 1974). Because robust schemes for estimating $N_i$ are still under active development, $N_i$ has not been included in the IIR operational products thus far.

Following the release of the Version 4 (V4) CALIOP data products, a new version of the IIR Level 2 data products has been developed. Input data products are i) Version 2 IIR Level 1b products that integrate corrections of small but systematic seasonal biases that were observed in the northern hemisphere in Version 1 (Garnier et al., 2017, 2018), and ii) V4 CALIOP 5-km cloud layer and aerosol layer products. This new IIR version is named V4 after the CALIOP products. As for the V4 CALIOP products, ancillary atmospheric and surface data are from the Global Modeling and Assimilation Office (GMAO) Modern-Era Retrospective analysis for Research and Applications, Version 2 (MERRA-2) model (Gelaro et al., 2017), and they replace the various versions of the GMAO Goddard Earth Observing System Version 5 (GEOS-5) model which were used in V3. The IIR V4 algorithm itself has been changed to improve both the estimates of effective emissivity derived over all surfaces and the subsequent microphysical indices retrievals. These improvements incorporate lessons learned from the combined analysis of numerous years of co-located V3 IIR and CALIOP level 2 data (G12; G13; G15). Ice clouds LUTs have been updated in V4 using state-of-the-art ice crystal single scattering properties (Bi and Yang, 2017), and V4 also includes independent retrievals using new parameterizations inferred from in situ measurements (M18). In response to the growing importance of better characterization of liquid water clouds for climate studies, V4 further takes advantage of improvements in microphysical indices to include specific retrievals of water cloud droplet size and liquid water path using dedicated LUTs.

This first paper (Part I) presents the main changes implemented in the V4 IIR Level 2 algorithm and describes improvements with respect to V3. All the changes implemented in V4 relate to the track algorithm. The algorithm used to extend the track retrievals to the IIR swath is as reported in G12 and therefore its description is not repeated here. Microphysical retrievals over oceans and comparisons with other A-Train retrievals will be presented in a companion "Part II" paper (Garnier et al., 2020). V4 retrievals over land, snow or sea ice with a specific emphasis on the changes in the surface emissivity will be presented in a forthcoming publication. The paper is organized as follows. The main updates to the scene classification algorithm are presented in Sect. 2. Section 3 describes the changes implemented to compute the effective emissivities in each IIR channel. The changes in the microphysical algorithm are detailed in Sect. 4 (effective diameter) and Sect. 5 (ice or liquid water path). Section 6 discusses how to estimate ice crystal and water droplet concentrations from the V4 CALIOP and IIR Level 2 products. The paper ends with a summary and concluding remarks in Sect. 7.

## 2 Scene classification

Both in V4 and in V3, the first task of the IIR algorithm is to classify the pixels in the scenes being viewed. This scene classification is based on the characteristics of the layers reported in the CALIOP 5-km cloud and aerosol products for layers detected by the CALIOP algorithm at 5-km and 20-km horizontal averaging intervals (Vaughan et al., 2009). This classification is designed to identify suitable scenes containing the required information for effective emissivity retrievals. The primary information provided by CALIOP includes the number of layers detected, their altitudes, types (i.e., cloud or aerosol), and mean volume depolarization ratio, and a determination of the opacity of the lowermost layer. The V4 classification algorithm is for the most part identical to V3 (G12), and only the main changes implemented in V4 are highlighted here.

For scenes that contain at least one cloud layer, the presence of lower semi-transparent aerosol layers is identified in the data products using the 'type of scene' parameter, but these aerosol layers are ignored when computing the emissivity of the (potentially multi-layered) cloud system. The rationale is that unless these low layers are dust (or volcanic ash) layers of sufficient optical depth, absorption in the IIR channels is negligible. In contrast, semi-transparent aerosol layers located above the cloud layer(s) are not ignored because they are more likely to be absorbing layers. These layers are those classified by CALIOP as smoke, volcanic ash, dust, or polar stratospheric aerosol (Kim et al., 2018).

Of importance for IIR passive observations, cloud layers with top altitudes lower than 4 km that were detected by CALIOP at single-shot resolution are cleared from the 5-km layer product to improve the detection of aerosols at coarser spatial resolutions (Vaughan et al., 2005). In V3, these single-shot "cleared clouds" were not reported in the 5-km layer products and hence were ignored by the IIR algorithm. However, clouds detected at single shot resolution have large signal-to-noise ratios (SNR), indicating that their optical depth is likely large and that they actually should not be ignored. This single shot detection frequently occurs when the overlying signal attenuation is small enough to ensure sufficiently large SNR, which favors scenes containing overlying optically thin aerosol or cloud layers. In V4, these single-shot cleared clouds are reported in the CALIOP 5-km products (Vaughan et al., 2020) and the IIR algorithm is able to use this new piece of information. A "Was_Cleared_Flag_1km" parameter is now available in the V4 IIR product, which reports the number of CALIOP single-shot clouds in the atmospheric column seen by the 1-km IIR pixel that were cleared from the 5-km layer products. Furthermore, scenes that were seemingly cloud-free in V3 are split into multiple categories in V4. Cloud-free scenes in V4 are pristine and have no single shot cleared clouds, while new types have been introduced to identify scenes that are cloud-free according to the 5-km layer products, but have at least one cleared cloud in the column. No IIR retrievals are attempted for these new scene types.

A lot of other parameters characterizing the scenes are reported in the V4 IIR product. Among them are the number of layers in the cloud system, as well as an "Ice Water Flag" which informs the user about the phase of the cloud layers included in the system, as assigned by the V4 CALIOP Ice/Water phase algorithm (Avery et al., 2020). A companion "Quality Assessment" flag reports the mean confidence in the feature type (i.e., cloud or aerosol) classification (Liu et al., 2019) and in the phase assignment for these cloud layers. The product also includes the number of tropospheric dust layers and of stratospheric aerosols layers in the column and the mean confidence in the feature type classification. All the suitable scenes are processed regardless of the confidence in the classifications and phase assignments reported in the CALIOP products, so that the user can define customized filtering criteria adapted to specific research objectives.

## 3 Effective emissivity and microphysical indices

### 3.1 Retrieval equations and sensitivity analysis

Before discussing the flaws that motivated changes in V4, we recall the retrieval equation of the effective emissivity, $\varepsilon_{eff,k}$, in each IIR channel, k (Platt and Gambling, 1971; Platt, 1973; G12) :

$$\varepsilon_{eff,k} = \frac{R_{k,m} - R_{k,BG}}{R_{k,BB} - R_{k,BG}} \qquad (1)$$

where $R_{k,m}$ is the calibrated radiance measured in channel k reported in the IIR Level 1b product, $R_{k,BG}$ is the background radiance in channel k that would be observed at the top of the atmosphere (TOA) in the absence of the studied cloud system, and $R_{k,BB}$ is the TOA radiance (also noted $B_k(T)$) that would be observed if the cloud system were a blackbody source of radiative temperature T.. These three radiances can be converted into equivalent brightness temperatures, noted respectively as $T_{k,m}$, $T_{k,BG}$, and $T_{k,BB}$, using the relationships reported in Sect. 2.4 of Garnier et al. (2018).

For each channel k, the effective absorption optical depth, $\tau_{a,k}$, is derived from $\varepsilon_{eff,k}$ as:

$$\tau_{a,k} = -\ln\left(1 - \varepsilon_{eff,k}\right) \qquad (2)$$

Finally, the microphysical indices $\beta_{eff}12/10$ and $\beta_{eff}12/08$ are written:

$$\beta_{eff}12/k = \frac{\tau_{a,12}}{\tau_{a,k}} = \frac{\ln\left(1 - \varepsilon_{eff,12}\right)}{\ln\left(1 - \varepsilon_{eff,k}\right)} \qquad (3)$$

The background and blackbody radiances are computed according to the scene classification introduced in Sect. 2.

The background radiance is determined either from the Earth's surface or, if the lowest of at least two layers is opaque, by assuming that this lowest layer behaves as a blackbody source. In both cases, the background radiance is preferably derived directly from relevant neighboring observations if they can be found. Otherwise, it is derived from computations using the FASt RADiative transfer model (FASRAD) (Dubuisson et al., 2005) and the meteorological and surface data available at global scale from meteorological analyses (MERRA-2 in V4). FASRAD calculations of the background radiance is required for ~75 % of all retrievals.

The blackbody radiance is computed using the FASRAD model and the estimated radiative temperature, which, in V3, is the temperature, $T_c$, at the centroid altitude, $Z_c$, of the 532-nm attenuated backscatter of the cloud system derived using interpolated temperature profiles. For multi-layer cases, the IIR algorithm computes an equivalent centroid altitude, and thereby sees the cloud system as a single layer.

Sensitivity of the retrieved quantities to errors in $T_{k,m}$, $T_{k,BG}$, and $T_{k,BB}$ have been discussed in details in G12, G13, and G15, and equations are repeated in Appendix A. Assuming no biases in the Version 2 calibrated radiances (Garnier et al., 2018), errors in $T_{k,m}$ and in $T_{k,BG}$ when the latter is derived from neighboring pixels are random, and equal to 0.15-0.3 K (G12). In contrast, errors in computed $T_{k,BG}$ and in $T_{k,BB}$ are composed of both systematic and random errors. Random errors in $T_{k,BG}$ from ocean surface computations were assigned after examining the distributions of the differences between observations and computations in clear sky conditions. In V4, the assigned random error is $\Delta T_{BG} = \pm 1$ K for all channels, which will be justified later in the paper. The assigned random error in $T_{k,BB}$ is $\Delta T_{BB} = \pm 2$ K for all channels to reflect uncertainties in the temperature profiles.

Random errors can be mitigated by accumulating a sufficient number of individual retrievals. However, systematic biases will remain and need to be reduced to the best of our ability. As a quantitative illustration, Fig. 1 shows the sensitivity of (a) $\varepsilon_{eff,12}$, (b) the inter-channel effective emissivity differences, noted $\Delta\varepsilon_{eff}12$-k, and (c) $\beta_{eff}12/k$ to systematic biases in $T_{BG}$ and $T_{BB}$ simulated using $T_{BG} = 285$ K and $T_{BB} = 225$ K. The sensitivities are inversely proportional to the radiative contrast between the surface and

the cloud. Thus, they are typically smaller in ice clouds, for which the temperature contrast over oceans is typically 60 K, as chosen in this example, than in water clouds that are closer to the surface. The black curves in Fig. 1 illustrate the impact of an identical bias of $dT_{12,BG} = dT_{k,BG} = +1$ K in all the channels. This positive bias increases $\varepsilon_{eff,12} \sim 0$ by $\sim 0.02$ and has an insignificant impact at $\varepsilon_{eff,12} \sim 1$. Even though the temperature bias is the same in all channels, $\Delta\varepsilon_{eff}$12-k and $\beta_{eff}$12/k are also impacted: at $\varepsilon_{eff,12} \sim 0.1$ (or optical depth $\sim 0.2$, corresponding to a thin cirrus cloud), $\beta_{eff}$12/10 (dashed line) is decreased by 0.03 and $\beta_{eff}$12/08 (dashed

dotted line) by 0.06. The red curves illustrate the impact of a channel-dependent bias in $T_{BG}$, by taking $dT_{12,BG} = 0$ K and $dT_{10,BG} = dT_{08,BG} = +0.1$ K. This modest inter-channel bias of $dT_{12,BG} - dT_{k,BG} = -0.1$ K induces a similar impact for both pairs of channels, with $\Delta\varepsilon_{eff}$12-k $\sim -0.002$ at $\varepsilon_{eff,12} \sim 0$ and $\beta_{eff}$12/k reduced by about 0.025 at $\varepsilon_{eff,12} \sim 0.1$. Finally, the blue curves are obtained by taking an identical bias in all channels of $dT_{12,BB} = dT_{k,BB} = +1$ K. This increases $\varepsilon_{eff,12} \sim 1$ by $\sim 0.01$ and has a negligible impact at $\varepsilon_{eff,12} \sim 0$. Again, this identical bias in all the channels changes $\Delta\varepsilon_{eff}$12-k and $\beta_{eff}$12/k: at $\varepsilon_{eff,12} = 0.95$, $\beta_{eff}$12/10 is increased by

0.02 and $\beta_{eff}$12/08 by 0.03. As seen in Fig. 1c, an acceptable bias of for instance 0.02 defines an emissivity domain of analysis ranging from 0.3 to 0.9. This domain is mostly limited in the low emissivity range, and refinements are necessary to extend this domain as much as possible.

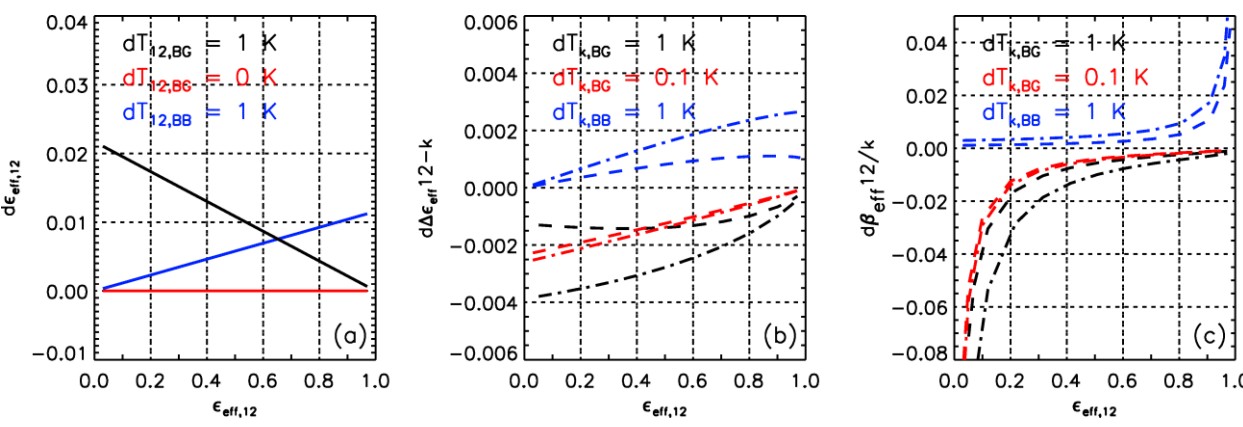

**Figure 1: (a) Sensitivity of $\varepsilon_{eff,12}$ to systematic errors $dT_{12,BG} = +1$ K (black), $dT_{12,BG} = 0$ K (red, no error), and $dT_{12,BB} = +1$ K (blue); (b)**
**sensitivity of $\Delta\varepsilon_{eff}$12-10 (dashed lines) and $\Delta\varepsilon_{eff}$12-08 (dashed dotted lines) to systematic errors $dT_{12,BG} = dT_{10,BG} = dT_{08,BG} = 1$ K (black), $dT_{12,BG} = 0$ and $dT_{10,BG} = dT_{08,BG} = 0.1$ K (red), and $dT_{12,BB} = dT_{10,BB} = dT_{08,BB} = 1$ K (blue); (c): same as (b), but for $\beta_{eff}$12/10 (dashed lines) and $\beta_{eff}$12/08 (dashed dotted lines). Simulations using $T_{BG} = 285$ K and $T_{BB} = 225$ K, and $\beta_{eff}$12/k=1.1.**

### 3.2 Motivations for changes in V4

    Changes in V4 were motivated by the need to reduce systematic errors in V3 microphysical retrievals that were made evident from
statistical analyses of the IIR V3 products. Because the sensitivity of the split-window technique decreases as effective emissivity approaches 0 and 1, $\Delta\varepsilon_{eff}$12-k is supposed to tend towards zero on average when $\varepsilon_{eff,12}$ tends towards 0 and towards 1. Examining whether this behavior was observed in our retrievals allowed us to identify errors related to the determination of background radiances when $\varepsilon_{eff,12}$ tended towards 0 and of blackbody radiances when $\varepsilon_{eff,12}$ tended towards 1 (G13). These tests were paired with comparisons between observed and modeled brightness temperatures, whenever relevant.

### 3.2.1 V3 biases at small emissivity

    Emissivity retrievals using $R_{k,BG}$ observed in neighboring pixels are a priori more robust than when this radiance is computed using a model. As discussed in G13, no biases were detected in V3 in the former case. However, when the ocean surface background radiances were computed using the model, median $\Delta\varepsilon_{eff}$12-k at $\varepsilon_{eff,12} \sim 0$ was clearly negative, down to $\sim -0.015$ for the 12-08 pair, which translated into significant low biases of the ice clouds microphysical indices at small emissivity (see Fig. 5 of G13 and Fig.
1c). This was due to channel-dependent biases in the computed radiances, which could be assessed independently by comparing

observations and computations in clear sky conditions. Consequently, the modeling of Earth surface radiance has been revisited in V4, as presented and evaluated in Sect. 3.3.

### 3.2.2 V3 biases at large emissivity

Large emissivities are typically found in so-called opaque clouds that fully attenuate the CALIOP signal. Importantly, "opaque" means opaque to CALIOP, that is cloud visible optical depth typically larger than 3 in V4 (Young et al., 2018) or effective emissivity expected to be larger than about 0.8. In opaque ice clouds, V3 $\varepsilon_{eff,12}$ was rarely larger than 0.95 (G12) and median $\Delta\varepsilon_{eff}12\text{-k}$ was minimum around $\varepsilon_{eff,12} = 0.95$ rather than 1. Both suggested that $\varepsilon_{eff,12}$ was systematically too small, and therefore that the cloud radiative temperature was underestimated. In other words, observations in opaque ice clouds tended to be warmer than the computed blackbody temperatures, by about 5 K (see Fig. 8 in G12), while this systematic positive bias was not observed for warm opaque water clouds. A similar contrast between ice and water clouds was also reported by Hu et al. (2010) when comparing IIR observations and mid-cloud temperatures. Stubenrauch et al. (2010) reported that for high opaque ice clouds, the radiative height determined by the Atmospheric Infrared Sounder (AIRS) on board the Aqua satellite is on average lower than the altitude of the maximum CALIOP 532 nm attenuated backscatter by about 10 to 20 % of the CALIOP apparent thickness. The warm bias between radiative temperature ($T_r$) and the centroid temperature $T_c$ used in V3 was explained theoretically in G15. The $T_r$ - $T_c$ difference was found between 0 and + 8 K for semi-transparent single-layered clouds and increased with cloud emissivity and geometric thickness, in agreement with previous studies (Stubenrauch et al., 2013 and references therein). Underestimating $T_r$ (and therefore TOA $T_{BB}$) yields under-estimates in $\varepsilon_{eff,12}$ and the microphysical indices. Note that Heidinger et al. (2010) infer cirrus radiative height from suitable pairs of channels using a range of expected values of $\beta_{eff}$ as a constraint. The problem here is reversed and is instead to estimate $T_r$ in order to infer microphysical indices. The determination of $T_r$ in ice clouds implemented in V4 is presented and discussed in Sect. 3.4.

### 3.3 Background radiance from ocean surface in V4

### 3.3.1 FASRAD model

The background radiance from the surface is computed using the FASRAD model fed by horizontally and temporally interpolated temperature, water vapor, and ozone profiles and skin temperatures. These ancillary data are from the MERRA-2 reanalysis products in V4. In V3, differences between observed and computed brightness temperatures (BTDoc) in clear sky conditions over oceans exhibited latitudinal and seasonal variations for all channels (G12), which appeared to be related to variations in the water vapor profiles near the surface to which the IIR window channels are the most sensitive. The water vapor absorption coefficients were updated in V4, to take advantage of the advances in atmospheric spectroscopy over the last decade (Rothman et al., 2013). Using MERRA-2 sea surface temperature and atmospheric profiles, the model was tuned to minimize the residual sensitivity of BTDoc to the column-integrated water vapor path (IWVP). This assessment was carried out in V4 pristine clear sky conditions; i.e., when no layers were detected anywhere in the column or if the column included only low semi-transparent non-dust aerosols in which no single-shot cleared clouds were detected within the IIR pixel (Sect. 2). Systematic biases remained for each channel, even at night where the clear sky mask is a priori the most accurate because of the increased CALIOP nighttime signal-to-noise ratio. Nighttime BTDoc was on average equal to -0.5 K at 08.65 µm, -0.35 K at 10.6 µm, and -0.2 K at 12.05 µm. These biases were explained by the combination of possible errors in the model, in the ancillary data, and in the calibration. We chose to reconcile observations and computations by using a new set of surface emissivity values (see Table 1) with no attempt to include surface temperature variations as reported from airborne measurements (Newman et al., 2005). The derived surface emissivity values used in V4 are close to 0.98 on average. It is noted that to save computation time, the contribution of the clear sky

downwelling radiance reflected by the surface is not included in the operational FASRAD model. Because the surface emissivity values are close to 1, the subsequent impact on their derived values is not significant.

Table 1: Surface emissivity over oceans in the three IIR channels in V3 and in V4.

|  | Channel 08.65 | Channel 10.6 | Channel 12.05 |
| --- | --- | --- | --- |
| Surface emissivity V3 | 0.9838 | 0.9906 | 0.9857 |
| Surface emissivity V4 | 0.971 | 0.984 | 0.982 |

As an illustration, median BTDoc is shown in Fig. 2a vs. IWVP derived from MERRA-2 for each IIR channel, both in V4 (solid lines) and in V3 (dashed lines). Over-plotted in green is the median MERRA-2 sea surface temperature ($T_s$). The results are shown for 6 months of nighttime data in 2006 (from July through December) between 60° S and 60° N to ensure that the dataset is not contaminated by sea ice. The number of clear sky IIR pixels used for this analysis is plotted in Fig. 2b. Even though the ancillary data are from GMAO GEOS 5.10 in V3 for this time period and from MERRA-2 in V4, the differences between V3 and V4 are mostly due to the changes in the radiative transfer model. The amplitude of the variations of median BTDoc with IWVP is drastically reduced in V4 compared to V3 and the inter-channel differences are significantly smaller. Between IWVP = 1 and 5 g.cm$^{-2}$, where most of the samples are found, V4 median BTDoc is between -0.2 and 0.2 K for the 3 channels. Using the V4 surface emissivities compensates for a residual 10-12 inter-channel BTDoc bias of -0.15 K and a residual bias of -0.3 K for the 08-12 pair. In contrast, the V3 median 10-12 and 08-12 inter-channels biases were up to - 0.7 K and – 1.8 K, respectively, at IWVP = 5 g.cm$^{-2}$.

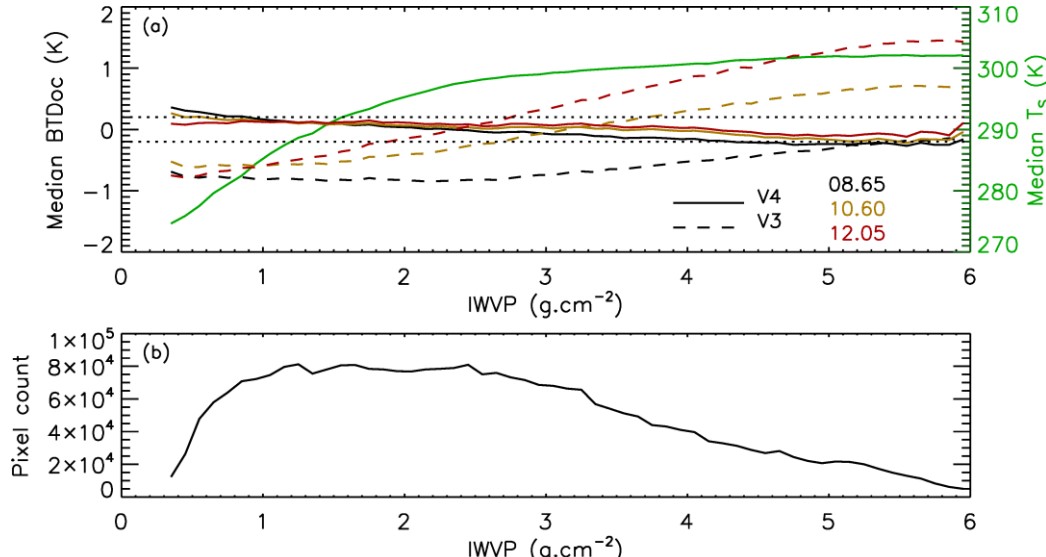

**Figure 2: (a): Median difference between observed and computed brightness temperatures (BTDoc) at 08.65 μm (black), 10.6 μm (brown), and 12.05 μm (red) vs. MERRA-2 IWVP in V4 pristine (no cleared clouds) nighttime clear sky conditions in V4 (solid lines) and in V3 (dashed lines) over oceans between 60° S and 60° N from July through December 2006. The horizontal dotted lines denote the -0.2 K and +0.2 K limits. Over-plotted in green is the median MERRA-2 surface temperature; (b): number of IIR pixels.**

### 3.3.2 Evaluation vs. latitude and season

In order to assess the errors in the computed background radiances used in the effective emissivity retrievals ($R_{k,BG}$, see Eq. 1) and in the corresponding computed brightness temperatures ($T_{k,BG}$), we analyzed distributions of BTDoc for different latitudes and seasons. Figures 3 and 4 show probability density functions (PDFs) of BTDoc at 12.05 μm, noted BTDoc(12), and of the 10-12 and 08-12 inter-channel BTDoc differences, noted BTDoc(10-12) and BTDoc(08-12), respectively. The results are for two months

in opposite seasons, namely January 2008 (Fig. 3) and July 2006 (Fig. 4), with computations from V4 (solid lines) and from V3
(dashed lines). The data are split into four 30-degree latitude bands between 60° S and 60° N, for both night (blue) and day (red).
Statistics of the V4 differences (median, mean, standard deviation, and mean absolute deviation) are reported in Table 2 for the
four latitude bands and globally (i.e., 60° S-60° N).

BTDoc(12) is overall less latitude-dependent in V4 than in V3, owing to the reduced bias related to IWVP in V4, and the width of
the distributions is reduced. The V4 global standard deviations are similar for nighttime (0.8 K) and daytime (0.9 K) data. Mean
V4 BTDoc(12) is larger for daytime than nighttime data at any latitude, by 0.2 K on average. As mentioned earlier, the V4 clear
sky mask is expected to be more accurate at night than during the day. Undetected absorbing clouds would decrease the brightness
temperature of the observations and therefore BTDoc(12), and a larger fraction of undetected clouds for daytime data would yield
smaller daytime BTDoc(12) and not larger values as observed here. A similar finding was reported in Garnier et al. (2017) for both
IIR and MODIS, suggesting that these differences are not due to calibration issues. The computations used a different model,
namely the 4A-OP radiative transfer model (Scott and Chédin, 1981), and ancillary data were from the ERA-Interim re-analyses
(Dee et al., 2011). It is unclear whether the small but systematic day vs. night differences are due to the V4 clear sky mask or other
reasons.

Again, the inter-channel differences are drastically reduced in V4 compared to V3, especially for the 08-12 pair of channels. In
V4, the absolute values of the mean inter-channel differences are smaller than 0.1 K globally. The worst cases are in July 2006 at
30-60° N (Fig. 4), where mean BTDoc(10-12) and BTDoc(08-12) are equal to -0.15 K and -0.26 K, respectively.  The global
standard deviations are around 0.31-0.35 K, notably smaller than 0.8-0.9 K found for BTDoc(12), because common biases due to
errors in sea surface temperature cancel out. Keeping in mind that the random noise at warm temperature is 0.15-0.2 K (G12) in
each channel, the standard deviations around 0.31-0.35 K can be largely explained by the random noise in the observed
temperatures, which is estimated to be $0.2 - 0.3$ K. Thus, the analysis of these inter-channel distributions shows that the uncertainty
in computed $T_{k,BG}$ can be taken identical in all channels. Based on the standard deviations in BTDoc(12), the random error $\Delta T_{BG}$ is
set to the conservative value $\pm 1$ K for all channels.

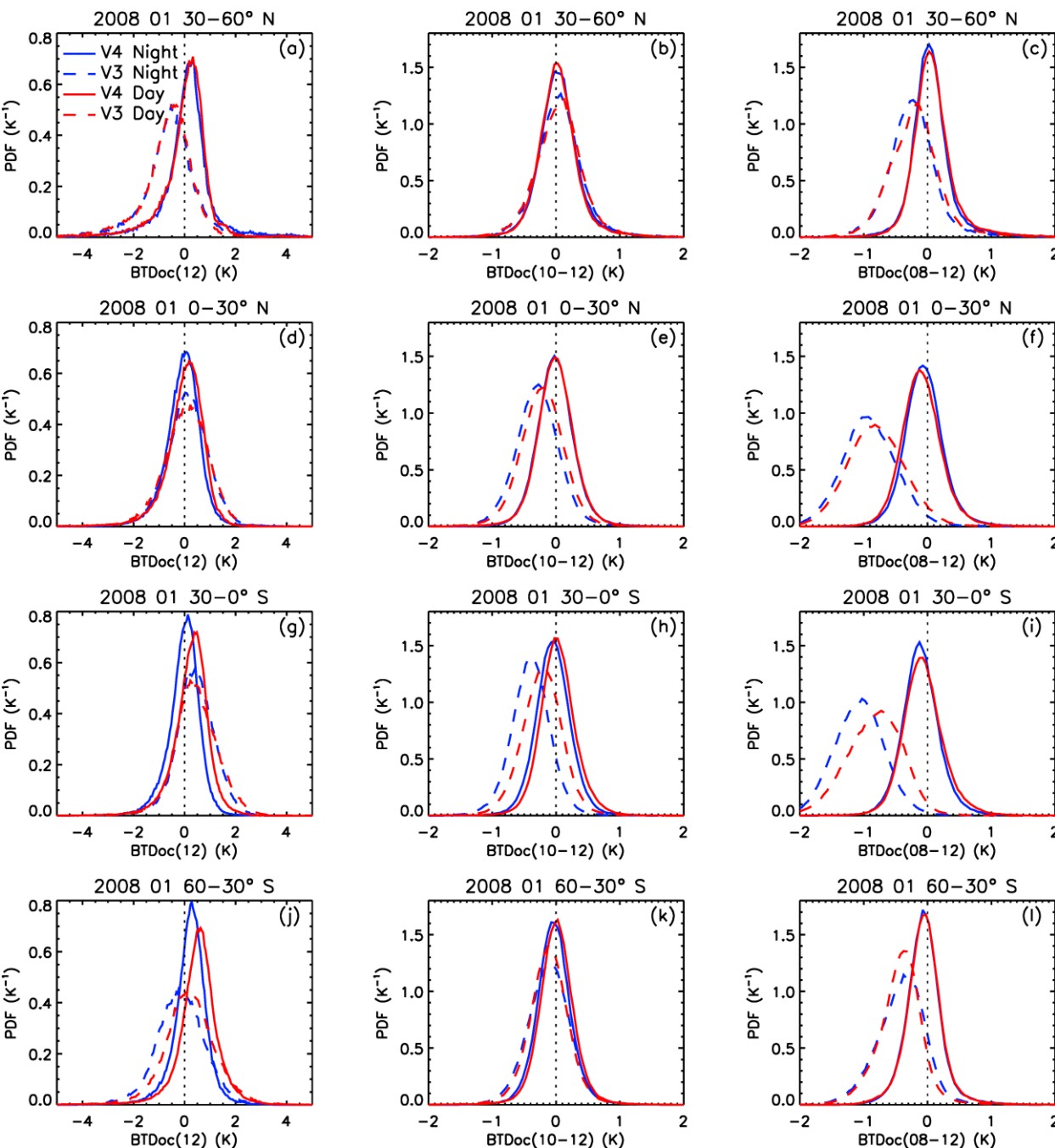

**Figure 3: Probability density functions (PDF) of the differences between observed and computed brightness temperatures (BTDoc) over oceans in January 2008 in V4 pristine (no cleared clouds) nighttime (blue) and daytime (red) clear sky conditions in V4 (solid lines) and in V3 (dashed lines). Panels a, d, g, j: BTDoc at 12.05 µm. Panels b, e, h, k: 10-12 inter-channel BTDoc difference. Panels c, f, i, l: 08-12 inter-channel BTDoc difference. The PDFs are shown at 30-60° N (panels a, b, c), 0-30° N (panels d, e, f), 30-0° S (panels g, h, i), and 60-30° S (panels j, k, l).**

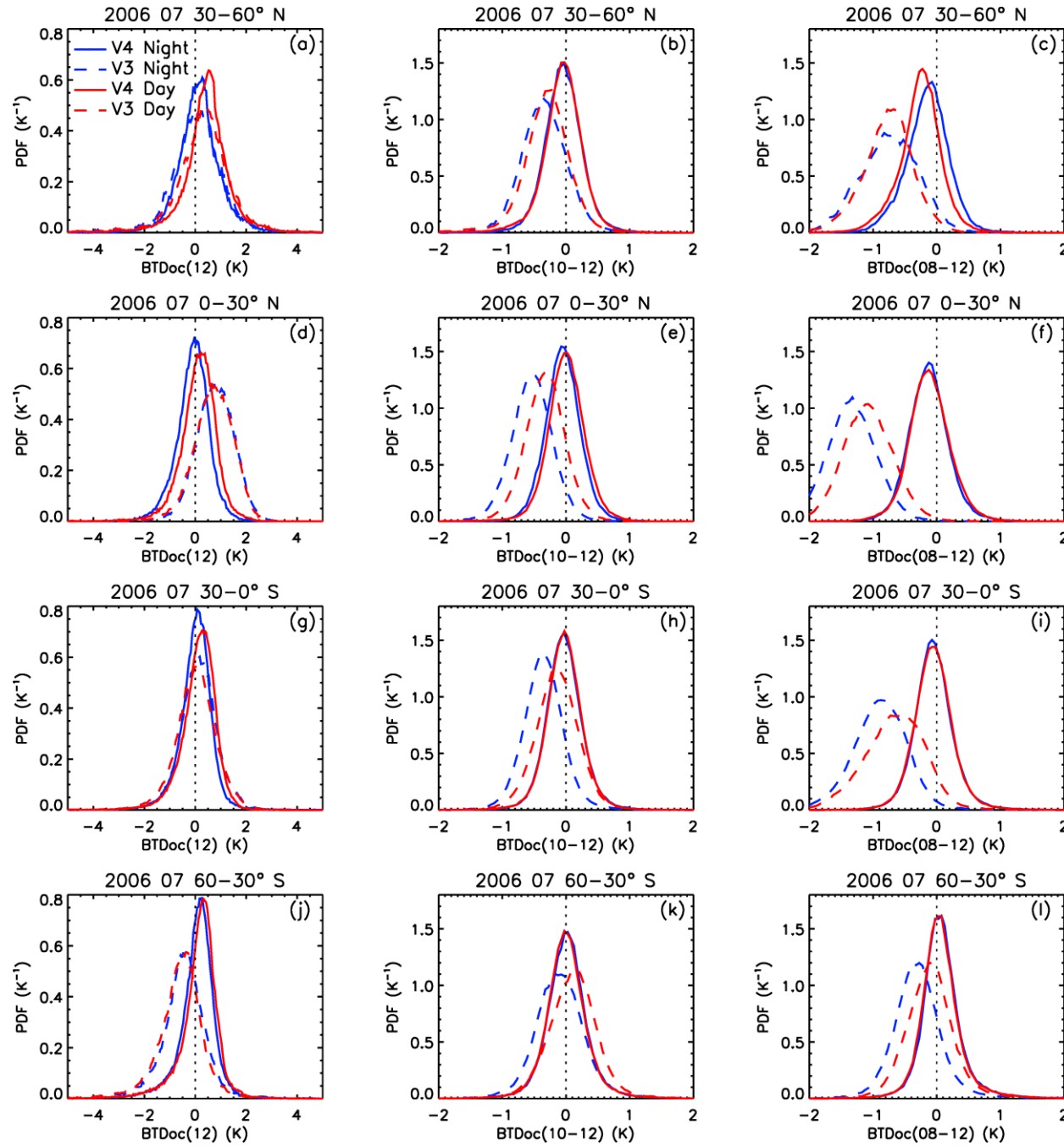

**Figure 4: Same as Fig. 3 but for July 2006.**

Table 2: V4 statistics (median, mean, standard deviation (STD), and mean absolute deviation (MAD)) of the differences between observed and computed brightness temperatures in V4 clear sky conditions (no cleared clouds) over oceans in January 2008 and in July 2006.

| January 2008 | # IIR pixels | | | BTDoc (12) (K) | | BTDoc (10-12) (K) | | BTDoc (08-12) (K) | |
|---|---|---|---|---|---|---|---|---|---|
| Latitude band | Night | Day | | Night | Day | Night | Day | Night | Day |
| 30-60° N | 63,523 | 77,381 | Median | 0.15 | 0.15 | 0.03 | 0.02 | 0.03 | 0.05 |
| | | | Mean | 0.19 | 0.00 | 0.04 | 0.03 | 0.07 | 0.09 |
| | | | STD | 1.58 | 1.35 | 0.37 | 0.37 | 0.38 | 0.39 |

| | # IIR pixels | | | BTDoc (12) | | BTDoc (10-12) | | BTDoc (08-12) | |
|---|---|---|---|---|---|---|---|---|---|
| | Night | Day | | Night | Day | Night | Day | Night | Day |
| | | | MAD | 0.81 | 0.73 | 0.25 | 0.25 | 0.24 | 0.25 |
| 0-30° N | 156,987 | 195,197 | Median | -0.02 | 0.09 | -0.01 | -0.01 | -0.06 | -0.10 |
| | | | Mean | -0.06 | 0.00 | -0.01 | -0.01 | -0.05 | -0.09 |
| | | | STD | 0.75 | 0.80 | 0.30 | 0.31 | 0.32 | 0.33 |
| | | | MAD | 0.53 | 0.57 | 0.23 | 0.23 | 0.24 | 0.25 |
| 0-30° S | 178,318 | 258,476 | Median | 0.02 | 0.31 | -0.05 | 0.02 | -0.11 | -0.08 |
| | | | Mean | -0.03 | 0.23 | -0.05 | 0.023 | -0.10 | -0.07 |
| | | | STD | 0.64 | 0.80 | 0.29 | 0.31 | 0.31 | 0.34 |
| | | | MAD | 0.47 | 0.54 | 0.22 | 0.22 | 0.23 | 0.25 |
| 30-60° S | 157,130 | 234,098 | Median | 0.26 | 0.56 | - 0.04 | 0.00 | - 0.06 | -0.06 |
| | | | Mean | 0.22 | 0.51 | - 0.04 | 0.01 | - 0.06 | -0.05 |
| | | | STD | 0.69 | 0.91 | 0.29 | 0.30 | 0.30 | 0.30 |
| | | | MAD | 0.49 | 0.59 | 0.21 | 0.22 | 0.21 | 0.22 |
| 60° S-60° N | 555,958 | 765,152 | Median | 0.09 | 0.31 | -0.03 | 0.01 | -0.07 | -0.06 |
| | | | Mean | 0.05 | 0.23 | -0.02 | 0.01 | -0.06 | -0.05 |
| | | | STD | 0.85 | 0.93 | 0.31 | 0.31 | 0.32 | 0.34 |
| | | | MAD | 0.54 | 0.60 | 0.23 | 0.23 | 0.23 | 0.24 |

| July 2006 | # IIR pixels | | | BTDoc (12) | | BTDoc (10-12) | | BTDoc (08-12) | |
|---|---|---|---|---|---|---|---|---|---|
| Latitude band | Night | Day | | Night | Day | Night | Day | Night | Day |
| 30-60° N | 52481 | 79,420 | Median | 0.15 | 0.46 | - 0.04 | -0.04 | -0.13 | -0.24 |
| | | | Mean | 0.19 | 0.38 | -0.04 | -0.06 | -0.15 | -0.26 |
| | | | STD | 0.97 | 1.27 | 0.31 | 0.35 | 0.35 | 0.34 |
| | | | MAD | 0.66 | 0.75 | 0.23 | 0.24 | 0.26 | 0.26 |
| 0-30° N | 71,144 | 103,625 | Median | -0.06 | 0.17 | -0.05 | 0.00 | - 0.13 | -0.12 |
| | | | Mean | -0.1 | 0.10 | -0.05 | 0.00 | -0.13 | -0.11 |
| | | | STD | 0.67 | 0.79 | 0.29 | 0.31 | 0.32 | 0.34 |
| | | | MAD | 0.5 | 0.55 | 0.22 | 0.23 | 0.25 | 0.26 |
| 0-30° S | 169,803 | 213,552 | Median | 0.06 | 0.20 | -0.03 | -0.03 | -0.06 | -0.05 |
| | | | Mean | 0.01 | 0.13 | -0.03 | -0.03 | -0.05 | -0.05 |
| | | | STD | 0.69 | 0.72 | 0.30 | 0.30 | 0.32 | 0.32 |
| | | | MAD | 0.48 | 0.51 | 0.22 | 0.22 | 0.24 | 0.24 |
| 30-60° S | 93935 | 108,760 | Median | 0.14 | 0.24 | 0.01 | 0.00 | 0.08 | 0.04 |
| | | | Mean | 0.06 | 0.14 | 0.02 | 0.01 | 0.08 | 0.06 |
| | | | STD | 0.83 | 0.82 | 0.36 | 0.36 | 0.36 | 0.35 |
| | | | MAD | 0.54 | 0.54 | 0.25 | 0.25 | 0.24 | 0.24 |
| 60° S-60° N | 387,363 | 505,357 | Median | 0.06 | 0.24 | -0.03 | -0.02 | -0.05 | -0.07 |
| | | | Mean | 0.03 | 0.16 | -0.02 | -0.02 | -0.05 | -0.07 |
| | | | STD | 0.77 | 0.87 | 0.32 | 0.33 | 0.35 | 0.35 |

| | | | MAD | 0.53 | 0.57 | 0.23 | 0.23 | 0.25 | 0.26 |
|---|---|---|---|---|---|---|---|---|---|

Again, the presence of clouds that were detected at single-shot resolution and later cleared from the 5-km layer product is forbidden in the V4 clear sky mask. The impact of this refinement in V4 is illustrated in Fig.5, which compares the BTDoc histograms in V4, in which single-shot clouds are specifically excluded, and in pseudo-clear sky conditions (i.e., which contain at least one single-shot cloud) over oceans between 60° S and 60° N in January 2008. When cleared clouds are present (light blue and orange), median and mean BTDoc(12) are smaller by 1.3 K and 2.2 K, respectively, and a marked negative tail down to about -8 K is observed, because these cleared clouds have a fairly large optical depth and are often colder than the surface. In this example, the fraction of IIR pixels that see at least one cleared cloud in the column is 35 % at night and 22 % for daytime data. The larger nighttime fraction is likely related to the fact that the probability for CALIOP to detect a cloud at single-shot resolution is larger at night due to the larger daytime background noise, so that the probability that these clouds are cleared from the product is larger at night. The mean and median values of the inter-channel BTDoc are barely impacted, showing that the cleared clouds induce a similar bias in the three IIR channels.

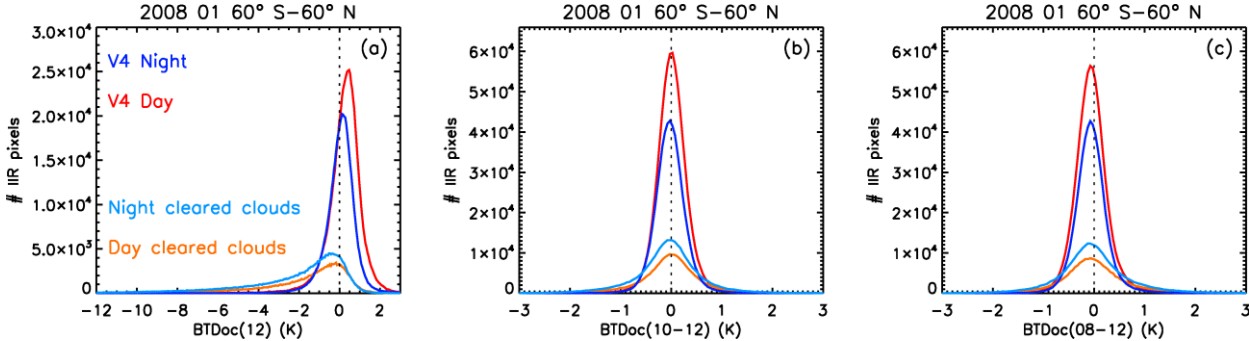

**Figure 5: Histograms of the differences between observed and computed brightness temperatures over oceans between 60° S and 60° N in January 2008 in V4 clear sky conditions (no cleared clouds) (navy blue: night; red: day) and in pseudo clear sky conditions (cleared clouds in the column) (light blue: night; orange: day). (a): BTDoc at 12.05 µm; (b) 10-12 and (c) 08-12 inter-channel BTDoc differences.**

### 3.4 Radiative temperature in V4

#### 3.4.1 Centroid altitude and temperature

Both in V3 and in V4, the first step into the computation of the radiative temperature is to determine the centroid altitude, $Z_c$, of the cloud system. The centroid altitude of each layer is reported in the CALIOP 5-km layer product, together with the 532-nm integrated attenuated backscatter (hereafter IAB) of each layer. IAB is corrected for the molecular contribution and for the attenuation resulting from the overlying layers, noted $T^2_{overlying}$. Following the rationale presented in Appendix B, the centroid altitude of a multi-layer cloud system composed of N layers is computed as:

$$Z_c = \frac{\sum_{l=1}^{l=N} Z_c(l) \cdot IAB(l) \cdot T^2_{overlying}(l)}{\sum_{l=1}^{l=N} IAB(l) \cdot T^2_{overlying}(l)} \tag{4}$$

For single-layer cases (N = 1), $Z_c$ is obviously the centroid altitude reported in the CALIOP data product. For multi-layer cases, the cloud system is seen as an equivalent single layer characterized by $Z_c$ given in Eq. (4), whose top and base altitudes are the top of the uppermost layer and the base of the lowermost layer, respectively. The approach is the same as in V3, except that, because of an error in the computation of $T^2_{overlying}$ in the V3 IIR algorithm, estimates of $Z_c$ could be too low by up to several kilometers in V3 multi-layer cases.

In V3, the radiative temperature ($T_r$) was set to the centroid temperature ($T_c$) for any cloud system. The approach is the same in V4, except when all the layers are classified as ice by the V4 ice/water phase algorithm (Avery et al., 2020). In the latter case, $T_r$ is derived from $T_c$ and parameterized functions, as presented and illustrated in the next section.

### 3.4.2 Radiative temperature in ice clouds

As demonstrated in G15, the radiative temperature $T_r(k)$ in channel k is the brightness temperature associated with the centroid radiance of the attenuated infrared emissivity profile within the cloud. For a cloud containing a number, n, of vertical bins, i, of resolution $\delta z$, with $i = 1$ to $i = n$ from base to top, this centroid radiance can be written as a function of radiance $R_k(i)$ of bin i and CALIOP particulate (i.e., cloud) extinction coefficient, $\alpha_{part}(i)$, as:

$$R_k = \frac{\sum_{i=1}^{i=n}\left(1-e^{-\left[\frac{\alpha_{part}(i)\cdot\delta z}{r}\right]}\right)\cdot R_k(i).e^{-\sum_{j=i+1}^{j=n+1}\left[\frac{\alpha_{part}(j)\cdot\delta z}{r}\right]}}{\varepsilon_{eff,k}} \tag{5}$$

The term $\alpha_{part}(i).\delta r/r$ in Eq. (5) is the absorption optical depth in bin i. The ratio, r, of CALIOP optical depth to IIR absorption optical depth is taken equal to 2 (G15). The radiance $R_k(i)$ is determined from the thermodynamic temperature in bin i.

On the other hand, $T_c$ is the temperature at the centroid altitude of the attenuated 532 nm backscatter coefficient profile, which is written as a function of altitude $Z(i)$ of bin i and $\alpha_{part}(i)$ as:

$$Z_c = \frac{\sum_{i=1}^{i=n}Z(i)\cdot\left(\beta_{part}(i)+\beta_{mol}(i)\right)\cdot e^{-2\sum_{j=i}^{j=n}\left[\eta\alpha_{part}(j)+\alpha_{mol}(j)\right]\delta z}}{\sum_{i=1}^{i=n}\left(\beta_{part}(i)+\beta_{mol}(i)\right)\cdot e^{-2\sum_{j=i}^{j=n}\left[\eta\alpha_{part}(j)+\alpha_{mol}(j)\right]\delta z}} \tag{6}$$

In Eq. (6), $\beta_{part}(i)$ is the cloud particulate backscatter in bin i, $\alpha_{mol}(i)$ and $\beta_{mol}(i)$ are the molecular extinction coefficient and backscatter, respectively, and $\eta$ is the ice cloud multiple scattering correction factor (Young et al. (2018) and references therein). Using V3 CALIOP extinction and backscatter profiles in semi-transparent ice clouds, the $T_r$-$T_c$ difference was found to increase with both cloud optical depth and geometric thickness (G15).

Because the CALIOP extinction profiles are not used in the IIR operational algorithm, the approach in V4 was to establish parameterized correction functions, $T_r(k)$ - $T_c$, for each channel k, and to correct the initial estimate $T_c$ that was used in V3 as $T_r(k) = T_c + [T_r(k)- T_c]$. These correction functions were derived off-line from the statistical analysis of a series of simulated extinction and attenuated backscatter profiles. In order to reproduce the variability associated to the various possible shapes of the extinction profiles, we chose to use actual V4 CALIOP profiles (8,000 profiles were used) rather than synthetic profiles. These initial CALIOP profiles were derived from single-layered semi-transparent clouds classified with high confidence as randomly oriented ice (ROI) by the V4 ice/water phase algorithm (Avery et al., 2020). Each CALIOP extinction (and backscatter) profile was scaled to simulate several pre-defined optical depths corresponding to several pre-defined effective emissivities using $r = 2$, and the attenuated backscatter profile was simulated by applying the required attenuation to the simulated total (molecular and particulate) backscatter profile. The simulations of $T_r(k)$ using Eq. (5) and of $T_c$ using Eq. (6) were carried out for $\varepsilon_{eff,k}$ ranging between 0.1 (or $\tau_{a,k} = 0.1$, see Eq. (2)) and 0.99 (or $\tau_{a,k} = 4.6$). Variations of $T_r(k)$ - $T_c$ with $\eta$ between 0.5 and 0.8 were also analyzed in order to cover the range of temperature-dependent values used in V4 (G15; Young et al., 2018). Variations with $\eta$ were not discussed in G15 because $\eta$ was taken constant and equal to 0.6 in V3.

The $T_r(k) - T_c$ differences were examined against the "thermal thickness" of the clouds; that is, the difference between the temperatures at cloud base ($T_{base}$) and at cloud top ($T_{top}$). Ninety percent of the CALIOP profiles used for this analysis had $T_{base} - T_{top}$ between 10 and 50 K. The median relative difference $(T_r - T_c)/(T_{base} - T_{top})$ was found to vary linearly with $T_{base} - T_{top}$, as illustrated in Fig. 6a for channel 12.05 µm using $\eta = 0.6$. Figures 6b and 6c show that the intercepts ($a_0$) and the slopes ($a_1$) of the regression lines vary with cloud absorption optical depth $\tau_{a,12}$. Furthermore, the $T_r - T_c$ differences increase with $\eta$ because the CALIOP signal is attenuated more quickly when less multiple scattering (i. e., larger $\eta$) contributes to the backscattered signal. As a result, both $a_0$ (Fig. 6b) and $a_1$ (Fig. 6c) increase as $\eta$ is increased from 0.5 to 0.8. Finally, the mathematical expression for the correction implemented in V4 is:

$$T_r(k) - T_c = a_0\left(\tau_{a,k}, \eta, k\right) \times \left[T_{base} - T_{top}\right] + a_1\left(\tau_{a,k}, \eta, k\right) \times \left[T_{base} - T_{top}\right]^2 \tag{7}$$

where the letter $k$ refers to the IIR channel. The corrections derived from Fig. 6a are shown in Fig. 6d. For a given value of $T_{base} - T_{top}$, $T_r - T_c$ increases with $\varepsilon_{eff,12}$ until $\varepsilon_{eff,12} = 0.7$-$0.8$ and is maximum for $\varepsilon_{eff,12} = 0.8$-$0.99$ (or $\tau_{a,12}$ between 1.6 and 4.6), where it represents 10 to 25 % of the cloud thermal thickness. We find that $T_r(k)$ is slightly larger at 10.6 µm than at 12.05 µm, by less than 0.3 K in the worst case, and somewhat larger at 08.65 µm than at 12.05 µm, but always by less than 1 K (not shown). Because at this stage of the algorithm, the final value of $\tau_{a,k}$ is still unknown, $\tau_{a,k}$ in Eq. (7) is the initial V3 value derived by taking $T_r = T_c$. No correction is applied when the initial emissivity is found larger than 1.

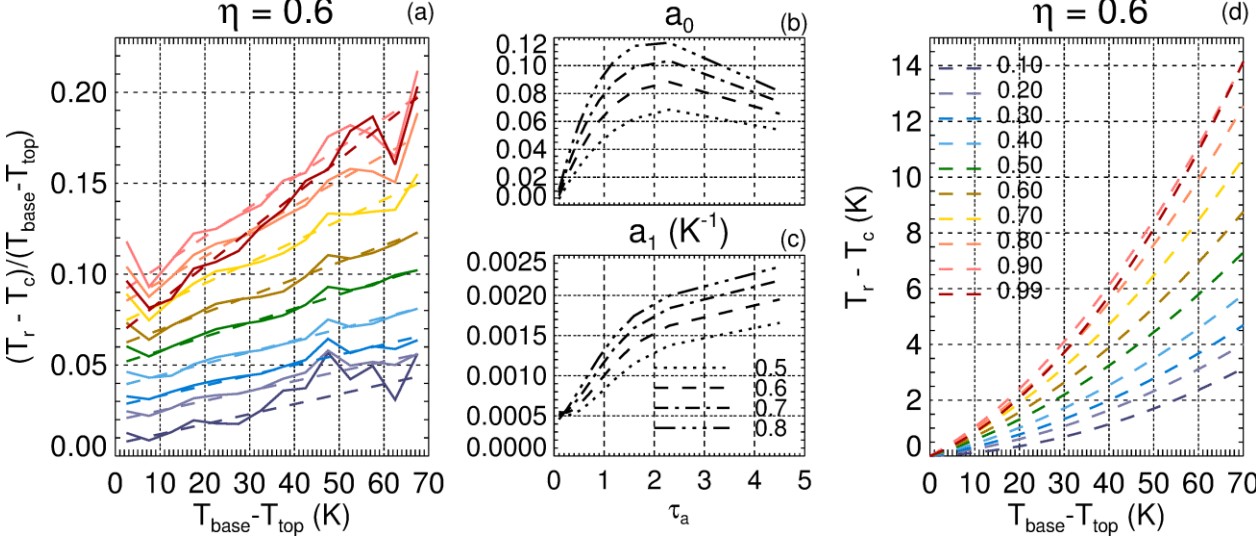

**Figure 6: V4 correction functions of ice cloud radiative temperature: (a) ($T_r$ - $T_c$)/($T_{base}$ - $T_{top}$) vs $T_{base}$ - $T_{top}$ for effective emissivities between 0.1 and 0.99 and $\eta$ = 0.6. The solid lines are median values from statistical analyses and the dashed lines are regression lines;(b) intercept and (c) slope of the regression lines vs. $\tau_{a,12}$ for $\eta$ between 0.5 and 0.8; (d) resulting V4 corrections $T_r$ - $T_c$ vs. $T_{base}$ - $T_{top}$ for effective emissivities between 0.1 and 0.99 and $\eta$ = 0.6. Channel 12.05 µm.**

The errors in the ice cloud radiative temperature corrections were assessed by comparing $T_r$ derived directly using the CALIOP extinction profiles with $T_r$ derived from Eq. (7). The statistics obtained from the same 8,000 CALIOP profiles as above are provided in Table 3, for both the $T_r - T_c$ correction and the correction error, for channel 12.05 µm. These statistics are provided for $\varepsilon_{eff,12}$ equal to 0.2, 0.6, and 0.99, and using $\eta$ equal to the extreme values 0.5 and 0.8. The median and mean correction errors are smaller than 0.25 K and significantly smaller than the median and mean corrections, which are found between 0.8 K and 5 K. The standard deviations of the correction errors are between 0.66 and 1.2 K at $\eta$ = 0.5 and between 0.7 and 1.75 K at $\eta$ = 0.8, while their mean

absolute deviations are smaller than 1.25 K. These quantities represent the estimated random error in the cloud radiative temperature correction resulting from the variability in the shape of the extinction profiles.

Table 3: Statistics (median, mean, standard deviation (STD), and mean absolute deviation (MAD)) of the $T_r$-$T_c$ correction at 12.05 µm and of correction errors for $\varepsilon_{eff,12}$ equal to 0.2, 0.6, and 0.99, using $\eta$ equal to 0.5 and 0.8. Channel 12.05 µm.

| | | $T_r$ -$T_c$ correction (K) | | | Correction error (K) | | |
|---|---|---|---|---|---|---|---|
| | | $\varepsilon_{eff,12} = 0.2$ | $\varepsilon_{eff,12} = 0.6$ | $\varepsilon_{eff,12} = 0.99$ | $\varepsilon_{eff,12} = 0.2$ | $\varepsilon_{eff,12} = 0.6$ | $\varepsilon_{eff,12} = 0.99$ |
| $\eta = 0.5$ | Median | 0.81 | 2.04 | 3.07 | 0.08 | -0.01 | 0.01 |
| | Mean | 0.92 | 2.22 | 3.43 | 0.1 | 0.02 | 0.13 |
| | STD | 0.55 | 1.16 | 1.98 | 0.66 | 0.75 | 1.2 |
| | MAD | 0.43 | 0.93 | 1.56 | 0.46 | 0.53 | 0.84 |
| $\eta = 0.8$ | Median | 1.32 | 3.8 | 4.50 | 0.17 | -0.01 | 0.01 |
| | Mean | 1.45 | 4.11 | 5.01 | 0.23 | 0.00 | 0.17 |
| | STD | 0.78 | 2.08 | 2.86 | 0.7 | 0.98 | 1.74 |
| | MAD | 0.62 | 1.65 | 2.26 | 0.50 | 0.70 | 1.24 |

Again, the maximum corrections are for clouds having initial effective emissivities in the 0.8-0.99 range and are similar for emissivities larger than 0.9. This range of initial emissivities is found for clouds that are opaque to CALIOP. It is noted that the corrections are a priori underestimated for opaque clouds. Because the CALIOP signal does not penetrate to the true base of opaque layers, the reported base is instead an apparent one, and so $T_{base}$ - $T_{top}$ is a priori too small. Figure 7 illustrates the impact of the correction applied to V4 opaque ice clouds classified as high confidence ROIs for IIR channel 12.05. The apparent thermal thickness (Fig. 7a) is larger at night (blue) compared to day (red), as already mentioned in Young et al. (2018). In this example, nighttime and daytime mean $\pm$ standard deviation of $T_{base}$ - $T_{top}$ are $28 \pm 13$ K and $21 \pm 9$ K, respectively. Similarly, the $T_r$ – $T_c$ corrections shown in Fig. 7b are larger at night. The discontinuities around $T_r$ – $T_c = 0$ in Fig. 7b are due to pixels with initial emissivity larger than 1 for which no correction is applied, which occurs more often at night. The smaller daytime apparent thickness is explained by the larger background noise in CALIOP daytime measurements, which increases the difficulty in accurately locating cloud boundaries. Consequently, both $T_c$ and $T_r$ are a priori more accurate at night. Figure 7c shows the $T_r$ - $T_{top}$ (solid lines) and $T_c$ - $T_{top}$ (dashed lines) differences relative to the apparent $T_{base}$ – $T_{top}$. After correction, the nighttime mean $\pm$ standard deviation of $(T_r$ - $T_{top})/(T_{base}$ - $T_{top})$ is $0.48 \pm 0.15$. This result is fully consistent with Stubenrauch et al. (2017), who report that the radiative cloud height derived from AIRS is, on average, at mid-distance between the CALIOP cloud top and cloud apparent base in high opaque clouds at night. Because the IR absorption above ice clouds is usually weak, $T_r$ is close to TOA $T_{BB}$. For reference, the dotted lines in Fig. 7c represent $(T_m$ - $T_{top})/(T_{base}$ - $T_{top})$, where $T_m$ is the measured brightness temperature (here $T_{12,m}$). $T_m$ represents the warmest possible value for $T_{BB} \sim T_r$ if all clouds had effective emissivity equal to unity. At night, $T_m$ is always located within the apparent cloud, at 61 % from the top as compared to 48 % for $T_r$. The $T_m$ - $T_r$ difference represents the maximum possible bias in the estimation of $T_r$ and is equal to 1.5 K on average. For daytime data, both $T_r$ and $T_m$ are lower in the apparent cloud than at night, and even below ($T_m > T_{base}$), which is at least in part due to the smaller daytime apparent thickness. Further evaluation will be carried out in the future using extinction profiles and true cloud base altitudes derived from the CloudSat radar.

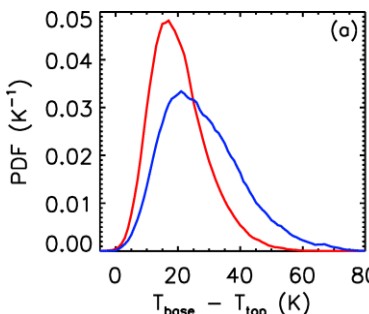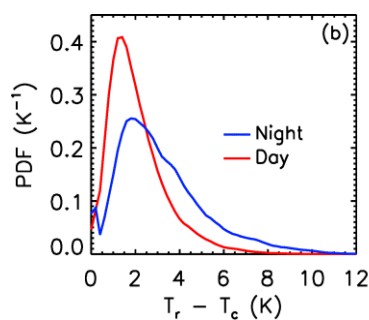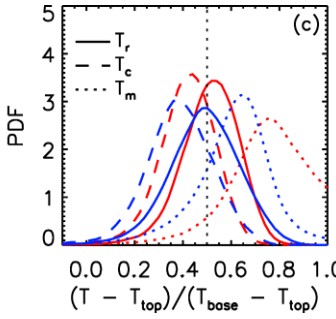

**Figure 7: Nighttime (blue) and daytime (red) probability density functions in V4 opaque ice clouds of (a) apparent thermal thickness $T_{base}$ - $T_{top}$, (b) $T_r$ - $T_c$ V4 correction, and (c) $(T - T_{top})/(T_{base} - T_{top})$ for $T = T_r$ (solid lines), $T = T_c$ (dashed lines), and $T = T_m$ (dotted lines) over oceans between 60° S and 60° N in January 2008. Channel 12.05 µm.**

With the introduction of corrections to the cloud radiative temperatures for ice clouds, V4 emissivities and microphysical indices now depend on the CALIOP ice/water phase classification. However, for optically very thin ice clouds, the corrections are typically smaller than 1 K, and, furthermore, these small corrections induce little changes in the final effective emissivity and microphysical indices (see Fig. 1). Thus, microphysical indices can be considered independent of the CALIOP ice/water phase at small emissivities, typically smaller than 0.3.

### 3.4.3 Radiative temperature in liquid water clouds

In case of opaque liquid water clouds, the observed brightness temperature ($T_m$) is, on average, close to the TOA $T_{BB}$ inferred from $T_c$. This indicates that the temperature at the centroid altitude is a good proxy for $T_r$, which is why no change was implemented in the V4 algorithm for liquid water clouds. Nevertheless, significant differences between V4 and V3 can arise from differences in the meteorological data. This is illustrated in Fig. 8, which shows PDFs of $T_m$ - $T_{BB}$ at 12.05 µm in V4 (solid lines) and in V3 (dashed lines) for opaque water clouds having identical centroid altitudes in V3 and in V4. These clouds are classified as water with high confidence by the V4 ice/water phase algorithm and they are the only layer detected in the column. The V3-V4 differences are due mainly to the different temperature profiles (GMAO GEOS 5.10 in V3 and MERRA-2 in V4), yielding different values of $T_c$ for an identical centroid altitude, and to a smaller extent to the changes in the water vapor profiles and in the FASRAD model. The V4 differences are -0.6 ± 2.2 K at night and 0.12 ± 2.7 during the day. The larger fraction of negative $T_m$ - $T_{BB}$ differences in V3, from - 2 K down to - 10 K, has been traced back to cases with strong temperature inversions near the top of the opaque cloud, which seem to be better reproduced in MERRA-2.

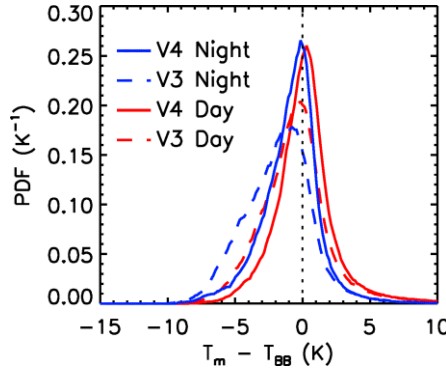

**Figure 8: Probability density functions of the V4 (solid) and V3 (dashed) nighttime (blue) and daytime (red) differences between measured brightness temperature and computed blackbody brightness temperatures over oceans between 60° S and 60° N in January 2008 for the IIR 12.05 µm channel.**

## 4 Effective diameter

The $\beta_{eff}12/k$ microphysical indices (Eq. 3) are interpreted in terms of ice crystal or liquid droplet effective diameter using LUTs built with the FASDOM model (Dubuisson et al., 2008) and available optical properties (G13). Following Foot (1988) and Mitchell et al. (2002), the effective diameter is defined as:

$$D_e = \frac{3}{2} \times \frac{V}{A} \tag{8}$$

where V and A are the total volume and the projected area that are integrated over the size distribution, respectively.

### 4.1 V4 Look-Up Tables

The difference between the V4 and the V3 ice LUTs is twofold: the ice habit models are different and a particle size distribution (PSD) is introduced in V4. Three ice habit models were used in V3. These were taken from the database described in Yang et al. (2005) and represented three families of relationships between $\beta_{eff}12/10$ and $\beta_{eff}12/08$: solid column, aggregate, and plate (G13). In practice, the plate model was rarely selected by the algorithm. In V4, the LUTs are computed using state-of-the-art ice crystals properties referred to as "TAMUice2016" by Bi and Yang (2017), which were updated with respect to "TAMUice2013" reported in 2013 (Yang et al., 2013). These optical properties determined by the Texas A&M University group are now widely used by the scientific community (Yang et al., 2018). Two models are used in V4: severely roughened "8-element column aggregate" (hereafter CO8) and "single hexagonal column" (hereafter SCO), for which the degree of the particle's surface roughness has little impact for the IIR channels. The former is the MODIS Collection 6 ice model for retrievals in the visible/near infrared spectral domain (except that the choice of the MODIS model was based on "TAMUice2013" properties), where the so-called "bulk" optical properties are computed using a gamma PSD with an effective variance of 0.1 (Hansen, 1971; Baum et al., 2011; Platnick et al., 2017). The same gamma PSD is chosen to compute the V4 IIR LUTs, whereas no PSD was introduced in V3 (G13).

For retrievals in liquid water clouds, which were added in V4, the LUTs are computed using the Lorenz-Mie theory with refractive indices from Hale and Querry (1973) and using the same PSD as for ice clouds.

As in V3, the LUTs are established for several values of $\varepsilon_{eff,12}$ (G13). Shown in Fig. 9 are the V4 (solid lines) and V3 (dashed lines) LUTs computed for $\varepsilon_{eff,12} = 0.23$ (visible optical depth ~ 0.5). This figure highlights that the $\beta_{eff}12/k$ microphysical indices are very sensitive to the presence of small particles in the PSD (Mitchell et al., 2010), with $\beta_{eff}12/k$ decreasing rapidly as $D_e$ increases up to 50 µm, and then tending asymptotically to ~ 1 at the upper limit of the sensitivity range; that is, $D_e = 120$ µm for ice crystals and 60 µm for liquid droplets. The retrieval of large particle sizes becomes very sensitive to noise and biases in the microphysical indices. In V4, the LUTs are extended to $D_e = 200$ µm for ice clouds and 100 µm for water clouds. Doing this allows the user to perform dedicated analyses when $\beta_{12/k}$ is only slightly smaller than the lower sensitivity limit, but $D_e$ retrievals beyond the sensitivity limit are very uncertain and flagged accordingly.

For ice clouds, the V4 $\beta_{eff}12/10$-$D_e$ relationships are relatively insensitive to the crystal model compared to the $\beta_{eff}12/08$-$D_e$ ones, owing to the larger single scattering albedo at 08.65 µm. The model dependence of the $\beta_{eff}12/10$-$\beta_{eff}12/08$ relationship is used as a piece of information about the ice model and the shape of the ice crystals to ultimately improve the $D_e$ retrievals. First, the algorithm identifies the model that provides the best agreement with the IIR parameters in terms of relationship between $\beta_{eff}12/10$ and $\beta_{eff}12/08$, and then $D_e$ is retrieved using this selected model. The water model is used for retrievals in liquid water clouds. For any model, $D_e$ is the mean of the effective diameters $D_e12/10$ and $D_e12/08$ when these two values can be retrieved from the respective $\beta_{eff}12/k$; i.e., $D_e = (D_e12/10 + D_e12/08)/2$. Both $D_e12/10$ and $D_e12/08$ are reported in the publicly distributed IIR data products.

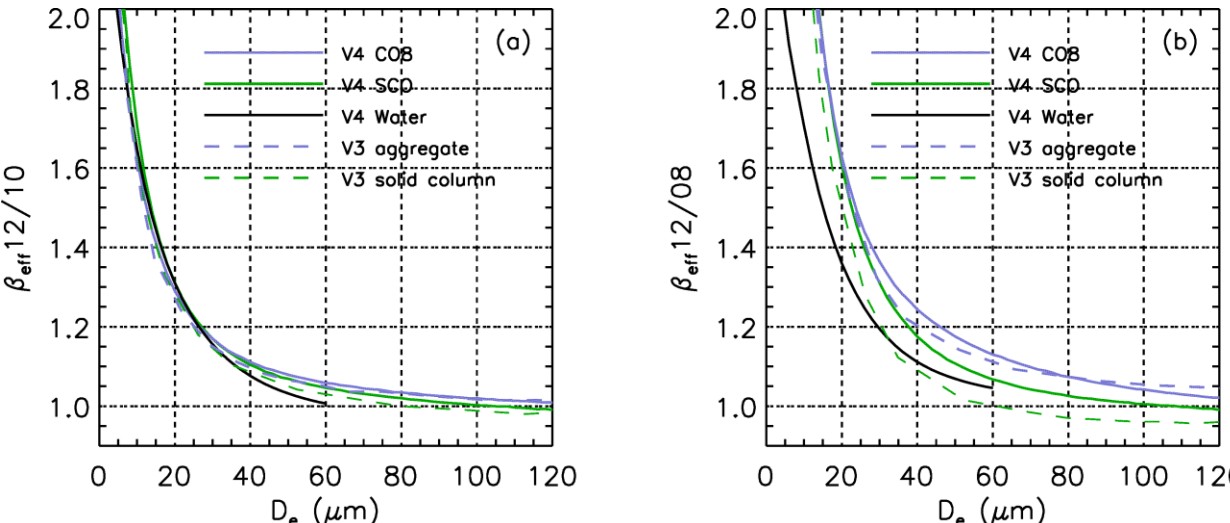

**Figure 9: (a) $\beta_{\text{eff}}12/10$ and (b) $\beta_{\text{eff}}12/08$ vs. $D_e$ using $\varepsilon_{\text{eff},12} = 0.23$ for the V4 LUTs (solid lines; purple: CO8, green: SCO, black: liquid water) and two V3 LUTs (dashed lines; purple: aggregate, green: solid column).**

### 4.2 Ice cloud model selection

As stated above, the ice cloud model is selected according to the relationship between $\beta_{\text{eff}}12/10$ and $\beta_{\text{eff}}12/08$. The theoretical

relationships derived from the V4 ice models are shown in Fig. 10, where the purple curves with square symbols show the CO8 model and the green curves with diamond symbols represent the SCO model. The colors of the symbols denote the value of $D_e$ between 20 μm and 120 μm. Because of the channel-dependent sensitivity to scattering, the overall relationship between $\beta_{\text{eff}}12/10$ and $\beta_{\text{eff}}12/08$ varies with effective emissivity, as seen when comparing the dotted ($\varepsilon_{\text{eff},12} = 0.1$) and dashed dotted ($\varepsilon_{\text{eff},12} = 0.9$) curves. Increasing $\varepsilon_{\text{eff},12}$ from 0.1 to 0.9 increases $\beta_{\text{eff}}12/10$ by only about 0.03 regardless of $D_e$, but tends to decrease $\beta_{\text{eff}}12/08$, by

up to 0.12 at $D_e = 20$ μm. As a result, the slope of the curves is increased by about 30 % from $\varepsilon_{\text{eff},12} = 0.1$ to 0.9 for both models, while the models themselves (purple and green curves) differ by only 10 % for a given $\varepsilon_{\text{eff},12}$ value, thereby showing the importance of properly taking scattering into account. For reference, the thin solid lines show the relationships derived from approximate $\beta_{\text{eff}}12/k$ defined by Parol et al. (1991) as:

$$\beta_{eff,proxy}12/k = \left[ Q_{12} \cdot \left(1 - \omega_{12} \cdot g_{12}\right) \right] / \left[ Q_k \cdot \left(1 - \omega_k \cdot g_k\right) \right] \tag{9}$$

where $Q_i$ is the extinction efficiency, $\omega_i$ is the single-scattering albedo, and $g_i$ is the asymmetry factor in the IIR channels $i = 12$ or $k$. For each crystal model, the approximate LUT value happens to be fairly close to the LUT value obtained at $\varepsilon_{\text{eff},12} = 0.9$.

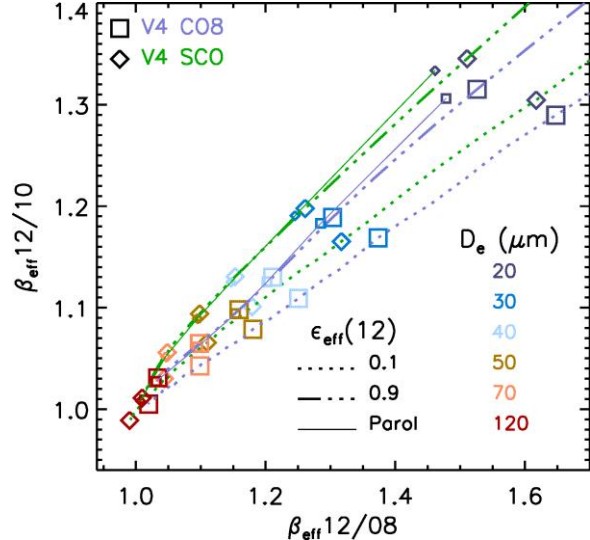

**Figure 10: V4 ice LUTs, showing β$_{eff}$12/08 (X-axis) vs. β$_{eff}$12/10 (Y-axis) for the CO8 model (squares and purple lines) and the SCO model (diamonds and green lines) for 6 values of D$_e$ between 20 and 120 μm. The LUT values are shown for ε$_{eff,12}$ = 0.1 (dotted lines) and = 0.9 (dashed dotted lines). The solid lines represent the approximate LUT values derived from Parol et al. (1991).**

### 4.3 Comparing the V3 and V4 ice models

In order to illustrate the impact of the changes introduced in the ice models in V4, Fig. 11 compares i) the V4 LUT values (solid lines and large symbols), ii) the V4 CO8 and SCO models but with no PSD (dashed dotted lines and small symbols), and iii) the V3 solid column and aggregate LUT values (dashed lines and different symbols), which had no PSD. For the three configurations, ε$_{eff,12}$ is arbitrarily taken equal to 0.23. First, we see that the V4 CO8 (purple solid) and the V3 aggregate (purple dashed) models are very similar in terms of relationship between β$_{eff}$12/10 and β$_{eff}$12/08. In contrast, V3 solid column (green dashed) appears to be systematically shifted towards smaller β$_{eff}$12/08 compared to V4 SCO (green solid). As a result, the difference between the V4 models is not as marked as the difference between the V3 models. Secondly, we note that for both V4 models, the solid and dashed dotted lines are very close, showing that the PSD chosen in V4 has a negligible impact on the relationship between β$_{eff}$12/10 and β$_{eff}$12/08, because it is quasi-linear. In other words, the ice model selection by the IIR algorithm is not impacted by the PSD introduced in V4. However, for a given D$_e$, β$_{eff}$12/10 and β$_{eff}$12/08 are larger with the V4 PSD (large symbols) than with no PSD (small symbols), because of the large sensitivity of β$_{eff}$12/k to the smallest crystals included in the distribution. In other words, including a PSD in V4 increases retrieved D$_e$.

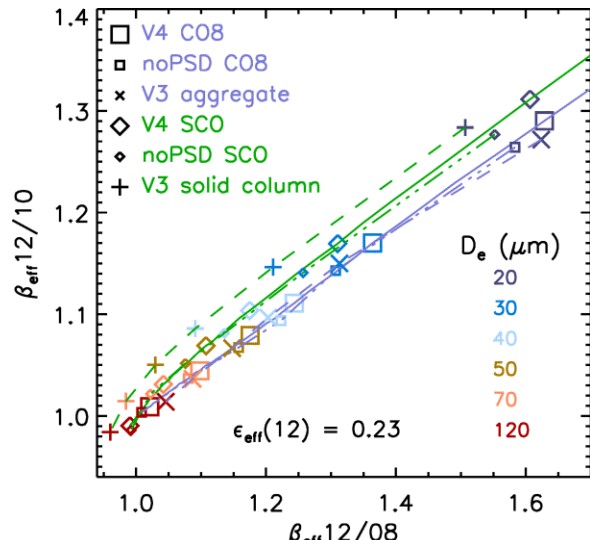

**Figure 11: The V4 ice LUT values (solid lines; CO8: large squares; SCO: large diamonds) are compared with the V4 LUT values with no PSD (dashed dotted lines and small symbols) and the V3 ice LUT values (dashed lines; cross: aggregate; plus signs: solid column), for 6 values of De between 20 and 120 μm and using $\varepsilon_{eff,12}$ = 0.23 for the 3 configurations.**

### 4.4 Comparing the V4 ice models and parameterizations from in situ observations

The four sets of analytical functions relating IIR $\beta_{eff}12/10$ and $D_e$ established in M18 were derived from in situ measurements in ice clouds performed during the SPARTICUS and TC4 field experiments, at mid-latitudes over land and at tropical latitudes over oceans, respectively. Because of uncertainties in the first bin, $N(D)_1$, of the measured size distributions (D < 15 μm), two LUTs were established for each campaign, one with $N(D)_1$ unmodified and the other with $N(D)_1$ set to zero to maximize the impact of a possible overestimate of $N(D)_1$. These four sets of $\beta_{eff}12/10 - D_e$ relationships are shown in Fig. 12, alongside the relationships derived from the V4 CO8 and the SCO models. For this comparison exercise, $\beta_{eff}12/10$ is computed using the approximate formulation given in Eq. 9.

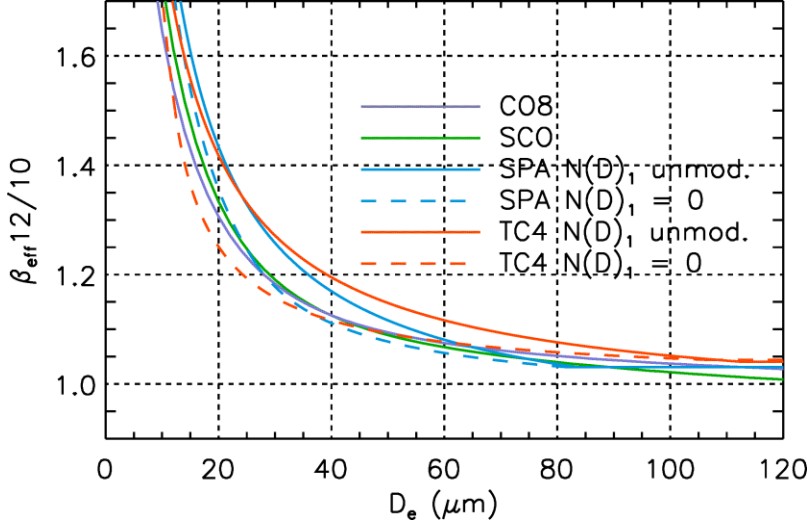

**Figure 12: $\beta_{eff}12/10$ vs. $D_e$ for the V4 LUTs (purple: CO8; green: SCO) and as derived by Mitchell et al. (2018) during the SPARTICUS (blue) and TC4 (red) field experiments using $N(D)_1$ unmodified (solid) or $N(D)_1$ = 0 (dashed).**

For a given $D_e$, $\beta_{eff}12/10$ is notably larger when $N(D)_1$ is not modified (blue and red solid lines) than when $N(D)_1$ is forced to zero (blue and red dashed lines), because the presence of small particles in the unmodified PSD increases $\beta_{eff}12/10$ more rapidly than $D_e$. The difference between the six $D_e$ values associated to a given value of $\beta_{eff}12/10$ is a measure of possible uncertainties resulting

from the LUTs. For instance, $\beta_{eff}12/10 = 1.6$ yields $D_e$ between 10 and 16 μm, and $\beta_{eff}12/10 = 1.1$ yields $D_e$ between 40 and 70 μm. The V4 SCO (green) and CO8 (purple) $\beta_{eff}12/10 – D_e$ relationships are fairly close to those derived using $N(D)_1 = 0$ (dashed), likely because the gamma functions with effective variance of 0.1 used in the V4 LUTs tend to fulfill this condition. Even though a PSD is now included for the computation of the V4 LUTs as an attempt to better simulate realistic conditions, the chosen gamma function is undoubtedly not adapted for any ice cloud globally.

## 5 Ice and liquid water path

### 5.1 Ice water path

As in V3, ice water path (IWP) in ice clouds is estimated from the visible extinction optical depth, $\tau_{vis}$, and $D_e$ using (Stephens, 1978; G13):

$$IWP = \frac{2}{3} \cdot \rho_i \cdot D_e \cdot \frac{\tau_{vis}}{Q_{e,vis}} \tag{10}$$

where $\rho_i$ is the ice bulk density ($\rho_i = 9.17 \times 10^2$ kg. m$^{-3}$) and $Q_{e,vis}$ is the visible extinction efficiency of the size distribution, typically close to 2. In V3, $\tau_{vis}$ was estimated from $\tau_{a,12}$ (Eq. 2) as:

$$\tau_{vis} = Q_{e,vis} \cdot \frac{\tau_{a,12}}{Q_{a,12}} \sim 2.\tau_{a,12} \tag{11a}$$

where $Q_{a,12}$ is the effective absorption efficiency at 12.05 μm of the size distribution, which was taken to be close to 1. However, as shown in Fig. 13a, the $\tau_{vis}/2\tau_{a,12}$ ratio varies with $D_e$, by up to 15 % for the V4 SCO model (green). In V4, $\tau_{vis}$ is estimated from $\tau_{a,12}$ and $\tau_{a,10}$ as:

$$\tau_{vis} \sim \tau_{a,12} + \tau_{a,10} \tag{11b}$$

As seen in Fig. 13b, using ($\tau_{a,12} + \tau_{a,10}$) instead of $2\tau_{a,12}$ as a proxy for $\tau_{vis}$ notably reduces $D_e$-dependent errors when $D_e$ is larger than 20 μm, as prevailingly found in ice clouds. The $\tau_{vis}/(\tau_{a,12} + \tau_{a,10})$ ratio slightly decreases as $\tau_{a,12}$ increases because of increasing influence of scattering in the effective infrared absorption optical depths, but by less than 5 % for the opaque clouds of $\tau_{a,12} = 3$ or $\varepsilon_{eff,12} = 0.95$. The overall errors in the V4 $\tau_{vis}$ estimates are within ± 6 % at $D_e = 20$ μm and ± 3 % at $D_e = 70$ μm. The simplified V4 formulation reduces the dependence on $D_e$ and is a straightforward approach to estimate $\tau_{vis}$ from IIR retrievals. It is convenient for comparisons with other sensors, providing that errors of about 5 % are acceptable.

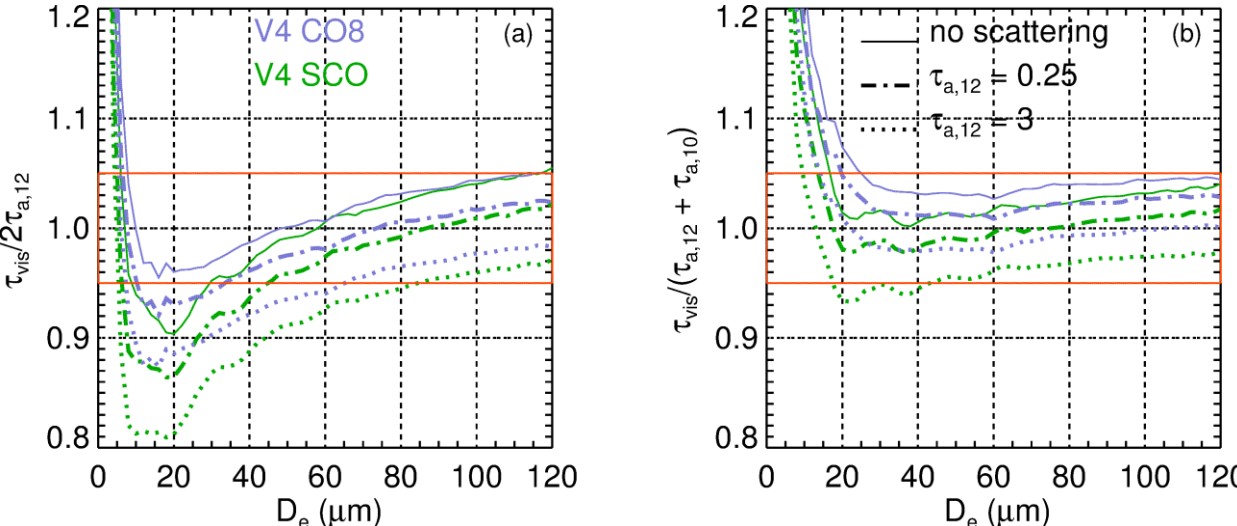

**Figure 13: Comparison of (a) $\tau_{vis}/2\tau_{a,12}$ and (b) $\tau_{vis}/(\tau_{a,12} + \tau_{a,10})$ vs. $D_e$ for the V4 CO8 (purple) and SCO (green) ice models. Simulations with no scattering (solid line), and with scattering using $\tau_{a,12} = 0.25$ (dashed dotted lines) and $\tau_{a,12} = 3$ (dotted lines). The red rectangle identifies the domain within $\pm 5$ % limits.**

### 5.2 Liquid water path

For liquid water clouds, liquid water path (LWP) is derived from $D_e$ and $\tau_{a,12}$ (Platt, 1976; Pinnick et al., 1979) as:

$$LWP = \frac{2}{3} \cdot \rho_w \cdot D_e \cdot \frac{\tau_{vis}}{Q_{e,vis}} = \frac{2}{3} \cdot \rho_w \cdot D_e \cdot \frac{\tau_{a,12}}{Q_{a,12}(D_e)} \tag{12}$$

where $\rho_w$ is the liquid water density. Unlike for ice clouds, the variation of $Q_{a,12}$ with $D_e$ is taken into account (Pinnick et al., 1979) and is represented using a fourth-degree polynomial for $D_e \leq 20$ µm, so that:

$$D_e \leq 20 \text{ µm} \qquad Q_{a,12}(D_e) = \sum_{i=0}^{i=4} a(i) \cdot D_e^{\ i} \tag{13a}$$

$$D_e > 20 \text{ µm} \qquad Q_{a,12}(D_e) = Q_{a,12}(20\mu m) \tag{13b}$$

In Eq. (13a), $D_e$ is in µm and the coefficients a(i) are reported in Table 4. In agreement with Pinnick et al. (1979), $Q_{a,12}$ increases quasi-linearly with $D_e < 10$ µm up to about 1, and then increases slowly up to 1.15 as $D_e$ increases from 10 to 20 µm. The polynomial function $Q_{a,12}(D_e)$ was established for $\tau_{a,12} = 0.25$ chosen to represent ST clouds. For opaque clouds of $\tau_{a,12} = 3$, $Q_{a,12}(D_e)$ is larger by only 5 %.

Table 4: Coefficients a(i), i=0,4 used in Eq. (13a) to compute LWP.

| | |
|---|---|
| a(0) | -0.102343 |
| a(1) | 0.236547 |
| a(2) | -0.0201336 |
| a(3) | 0.000859505 |
| a(4) | -0.0000144792 |

## 6 Particle concentration in ice and liquid clouds

Because IIR is a passive sensor, IIR Level 2 primary retrievals are vertically integrated quantities such as $\tau_{a,k}$ and IWP or LWP. Similarly, $D_e$ represents a layer average. Even though not provided in the V4 products, equivalent layer absorption coefficient and layer ice or water content can be derived for specific studies, and ultimately ice and liquid cloud concentrations can be determined.

### 6.1 Equivalent layer absorption coefficient and layer ice or liquid water content

The IIR retrievals are all tied to the retrieved effective emissivities. As demonstrated in G15, $\varepsilon_{eff,k}$ is the vertical integration of an attenuated effective emissivity profile, which can be determined from the CALIOP extinction profile, $\alpha_{part}(i)$. Looking at Eq. (5) used to derive the cloud radiative temperature and ultimately establish the correction functions presented in Sect. 3.4.2, we see that we can define an IIR weighting function, $WF_{IIR}(i)$, as:

$$WF_{IIR}(i) = \frac{1 - e^{-\left[\alpha_{part}(i) \cdot \delta z / r\right]}}{\varepsilon_{eff,k}} \cdot e^{-\sum_{j=i+1}^{j=n+1}\left[\alpha_{part}(j) \cdot \delta z / r\right]} \tag{14}$$

This applies to semi-transparent clouds whose true base is detected by CALIOP. This concept has been used in M18 to compute an equivalent effective thickness seen by IIR, $\Delta Z_{eq}$, derived from the geometric thickness, $\Delta Z$, as:

$$\frac{1}{\Delta Zeq} = \frac{1}{\Delta Z} \times \frac{1}{\tau_{vis}} \times \sum_{i=1}^{i=n} \alpha_{part}(i) . WF_{IIR}(i) . \delta z \tag{15}$$

$\Delta Z_{eq}$ was found equal to 30% to 90 % of $\Delta Z$ for ice clouds. The IIR equivalent layer absorption coefficient, $\alpha_{abs,eq}(k)$, is defined as:

$$\alpha_{abs,eq}(k) = \tau_{a,k} / \Delta Z_{eq} \tag{16}$$

Likewise, the IIR equivalent layer ice water content (IWC) or the IIR equivalent layer liquid water content (LWC) are written:

$$IWC(LWC) = IWP(LWP) / \Delta Z_{eq} \tag{17}$$

### 6.2 Ice crystal and water droplet concentration

First characterizations of particle concentrations have been developed for ice clouds. Following M18, the ice crystal concentration ($N_i$) in semi-transparent ice clouds can be derived as:

$$N_i = IWC \times \left(\frac{N_i}{IWC}\right)_{\beta_{eff} 12/10} \tag{18}$$

where the ($N_i$/IWC) ratio is a function of $\beta_{eff}12/10$ that was derived from in situ observations, depending on hypotheses in the measured PSD (see Table 1 in M18). As seen from M18, the uncertainty in the derivation of $N_i$/IWC increases rapidly as $\beta_{eff}12/10$ decreases below 1.1, e.g. as the effective diameter of ice crystals grows.

In the case of liquid water clouds, a similar approach can be used and the droplet concentration, $N_d$, can be written:

$$N_d = LWC \times \left( \frac{N_d}{LWC} \right)_{R_e} \tag{19}$$

Here, the ($N_d$/LWC) ratio can be written as a function of the effective radius $R_e = D_e/2$ as:

$$\frac{N_d}{LWC} = \frac{3}{4\pi} \times \frac{1}{\rho_w} \times \frac{1}{kR_e^3} \tag{20}$$

where k is a factor determined from the ratio of the mean volume radius and the effective radius. Two values k=0.67 and k=0.80 (with an uncertainty of 0.05) have been proposed for ocean and land respectively as derived from in situ measurements (Martin et al., 1994).

## 7 Summary and perspectives

The IIR Level 2 algorithm has been modified in the V4 data release to improve the accuracy of the microphysical indices in clouds of very small (close to 0) and very large (close to 1) effective emissivities. In addition, a new set of LUTs is used to retrieve ice cloud microphysical properties, and the retrievals have been extended to liquid water clouds.

Improving the microphysical indices at emissivities typically smaller than about 0.2 required improvements in the accuracy of the simulated background radiances. In this first of a pair of papers describing the V4 IIR level 2 updates, we focused on retrievals above the oceans. The changes in the new radiative transfer calculations using sea surface and atmospheric data from MERRA-2 were evaluated through comparisons with IIR measurements in clear sky conditions. These clear air conditions were identified using co-located CALIOP observations and were refined in V4 using additional information now reported in the V4 CALIOP 5 km layer products. Water vapor absorption and sea surface emissivities in each IIR channel were adjusted to reconcile observations and simulations. In V4, clear air observations and simulations agree within ± 0.2 K on average at night. The inter-channel 08-12 and 10-12 differences are drastically reduced from several tens of Kelvins in V3 to less than 0.1 K on average in V4.

The retrieval of cloud properties is increasingly difficult as effective emissivity approaches 1, because the sensitivity of the technique decreases when the measured radiance is close to the cloud equivalent blackbody radiance. Biases in the determined value of the blackbody brightness temperature thus have growing importance. In V3, we used the CALIOP centroid altitude as a proxy for the equivalent radiative altitude, but it was shown later that in case of ice clouds, the corresponding infrared radiative temperature was underestimated. We have minimized this bias in V4 by refining the relationship between lidar geometric altitude and infrared radiative temperature. We have implemented a parameterized correction of the V3 estimates which is a function of ice cloud thermal thickness, cloud absorption optical depth, and CALIOP multiple scattering correction factor. This correction is expected to both increase the number of valid retrievals of crystal sizes and reduce biases for ice clouds of large optical depth.

One of the specific features of the IIR algorithm is accounting for the relationships between the 12/10 and 12/08 microphysical indices in order to retrieve the particle effective diameter. New ice optical properties ("TAMUice 2016") have been used and two

crystal shapes have been selected to determine theoretical values of the microphysical indices as a function of effective diameter. One of these models is the 8-element column aggregate model selected by the MODIS science team for the Collection 6 products, and the other one is the single hexagonal column model. In V4, the bulk properties are computed using the same gamma PSD as selected by the MODIS team. The selection of the crystal model used for the retrievals is not impacted by this assumed size distribution, whereas introducing a size distribution in V4 increases the retrieved diameters. Another independent approach for deriving effective diameters is discussed, which relies on parameterizations based on in situ measurements from the SPARTICUS and TC4 field experiments used to determine the relationship between the 12/10 microphysical index and effective diameter. A simple and fairly accurate formulation of the ice cloud visible optical depth is proposed, which is based on IIR absorption optical depths at both 10.6 and 12.05 µm, and which could be conveniently used for comparisons with other sensors. This formulation is used to provide ice water path estimates.

The IIR V4 algorithm now includes a dedicated retrieval for liquid water clouds. These water clouds were not a priority in V3, due to the smaller radiative contrast between water clouds and the surface compared to ice clouds, and the resulting larger uncertainties. The water cloud retrieval was introduced in V4 because the uncertainties will be smaller than in V3, owing to the reduced biases, and because the IIR retrieval technique is well adapted for the smaller particle sizes found in liquid water clouds. Our primary targets will be supercooled liquid water clouds.

The changes and improvements in the V4 IIR Level 2 products resulting from the changes implemented in the V4 algorithm are presented in a companion (Part II) paper. One key feature of the IIR algorithm is the initial scene classification inferred from the co-located CALIOP observations, and specifically from the CALIOP 5-km cloud and aerosol layer products. The synergy with CALIOP could be reinforced by using the CALIOP extinction profiles to infer the in-cloud IIR weighting function and thus better characterize the fraction of the cloud layer to which IIR is sensitive, as was implemented for ice concentration retrievals in M18. This would improve the equivalent vertical resolution of the geophysical parameters retrieved by IIR and ultimately open the possibility to report vertically resolved parameters such as ice crystals and liquid droplet concentrations in a future version of the operational products.

## Appendix A: Sensitivity analysis and uncertainties

As seen from Eq. (1) and as discussed in G12, G13, and G15, the uncertainty in $\varepsilon_{eff,k}$ in each channel k includes three terms associated with the uncertainty in the measured radiance $R_{k,m}$, in the background radiance $R_{k,BG}$, and in the blackbody radiance $R_{k,BB}$. These three terms are inversely proportional to the radiative contrast, $R_{k,BG} - R_{k,BB}$. After defining $R'_{k,x}$, where the subscript x refers to m, BG or BB, as

$$R'_{k,x} = \frac{\partial R_{k,x}}{\partial T} \cdot \frac{1}{R_{k,BG} - R_{k,BB}} \tag{A1}$$

where T is the equivalent brightness temperature, the sensitivity $d\varepsilon_{k,x}$ of $\varepsilon_{k,x}$ to an error $dT_{k,x}$ in the brightness temperature that is equivalent to the radiance $R_{k,x}$ is

$$d\varepsilon_{k,m} = -R'_{k,m} \cdot dT_{k,m} \tag{A2a}$$

$$d\varepsilon_{k,BG} = \left(1 - \varepsilon_{eff,k}\right) R'_{k,BG} \cdot dT_{k,BG} \tag{A2b}$$

$$d\varepsilon_{k,BB} = \varepsilon_{eff,k} \cdot R'_{k,BB} \cdot dT_{k,BB}$$ (A2c)

The sensitivity of $\tau_{a,k}$ to an error $dT_{k,x}$ is

$$\left(d\tau_{a,k}\right)_x = \frac{d\varepsilon_{k,x}}{1-\varepsilon_{eff,k}}$$ (A3)

Finally, the relative sensitivity of $\beta_{eff}12/k$ to an error $dT_{k,x}$ is

$$\frac{\left(d\beta_{eff}12/k\right)_x}{\beta_{eff}12/k} = \frac{-d\varepsilon_{12,x}}{\left(1-\varepsilon_{eff,12}\right)\ln(1-\varepsilon_{eff,12})} + \frac{d\varepsilon_{k,x}}{\left(1-\varepsilon_{eff,k}\right)\ln(1-\varepsilon_{eff,k})}$$ (A4)

Equations A2a-c, A3, and A4 are used to compute the uncertainties $\Delta\varepsilon_{k,x}$, $\left(\Delta\tau_{a,k}\right)_x$, and $\left(\Delta\beta_{eff}12/k\right)_x$ associated to the uncertainties $\Delta T_{k,x}$, and the overall uncertainty is then estimated by assuming that these three uncertainty terms are not correlated. Computing $\left(\Delta\beta_{eff}12/k\right)_x$ requires establishing whether the two terms of Eq. (A4) are correlated. Assuming no bias in the calibration, the error $\Delta T_{k,m}$ represents the overall radiometric random noise for an individual pixel in for each channel k, and the

680 errors in the respective channels are not correlated. Regarding the background radiance, $\Delta T_{12,BG}$ and $\Delta T_{k,BG}$ depend on the way in which the background radiances are determined. If from neighboring pixels, $\Delta T_{12,BG}$ and $\Delta T_{k,BG}$ are due to the radiometric random noise and are not correlated. In contrast, when the background radiances are computed using the FASRAD model and the same ancillary data, $\Delta T_{12,BG}$ and $\Delta T_{k,BG}$ are correlated. Finally, because the blackbody radiances result from the cloud radiative temperature derived from $T_c$ and the $T_r$ - $T_c$ correction functions, $\Delta T_{12,BB}$ and $\Delta T_{k,BB}$ are also correlated.

**Appendix B: centroid altitude in multi-layer cloud systems**

The effective emissivity retrieval equation (Eq. 1) is valid regardless of the number of layers in the cloud system to be analyzed. For single-layer systems, the centroid altitude of the 532-nm attenuated backscatter, $Z_c$, is read directly in the 5-km CALIOP layer product. For multi-layer cloud systems, the IIR algorithm computes the equivalent centroid altitude of the cloud system, as presented below.

For a given layer, $Z_c$ is defined as (Vaughan et al., 2005):

$$Z_c = \frac{\int_{z_{top}}^{z_{base}} z(r) \cdot B_{532}(r) dr}{\int_{z_{top}}^{z_{base}} B_{532}(r) dr}$$ (B1)

where $B_{532}(r)$ is the 532-nm total attenuated backscatter coefficient at altitude z(r) corrected for the attenuation due to molecules and ozone. On the other hand, the 532-nm layer-integrated attenuated backscatter is

$$\gamma'_{532} = \int_{z_{top}}^{z_{base}} B_{532}(r)dr - dB_{532}$$ (B2)

where dB$_{532}$ represents the correction for the contribution from molecular scattering (Vaughan et al., 2005). In the 5-km layer product, the "Integrated_Attenuated_Backscatter_532" parameter (hereafter IAB) is $\gamma'_{532}$ corrected for the attenuation resulting from the overlying layers (https://www-calipso.larc.nasa.gov/resources/calipso_users_guide/data_summaries/layer/index_v420.php#integrated_attenuated_backscatter_532, last access 14 September 2020), which is noted $T^2_{overlying}$, so that:

$$\gamma'_{532} = IAB \cdot T^2_{overlying} \tag{B3}$$

For cloud layers of sufficient optical depth, the molecular contribution is weak compared to the particulate one, and the denominator in Eq. (B1) is approximatively $\gamma'_{532}$ or $IAB \cdot T^2_{overlying}$ , and the numerator is approximatively $Z_c \cdot IAB \cdot T^2_{overlying}$ . Assuming again that the contribution from molecular scattering can be neglected, the centroid altitude of a cloud system composed of N layers, $l$, is computed as:

$$Z_c = \frac{\sum_{l=1}^{l=N} \int_{z_{top(l)}}^{z_{base(l)}} z(r) \cdot B_{532}(r) dr}{\sum_{l=1}^{l=N} \int_{z_{top(l)}}^{z_{base(l)}} B_{532}(r) dr} = \frac{\sum_{l=1}^{l=N} Z_c(l) \cdot IAB(l) \cdot T^2_{overlying}(l)}{\sum_{l=1}^{l=N} IAB(l) \cdot T^2_{overlying}(l)} \tag{B4}$$

**Appendix C: Glossary**

| Notation | Description |
|---|---|
| $\alpha_{abs,eq}(k)$ | IIR equivalent absorption coefficient in channel k |
| $\alpha_{part}$ | CALIOP particulate extinction coefficient |
| BTDoc | Difference between observed and computed brightness temperatures in clear sky conditions, channel not specified |
| BTDoc(12) | Difference between observed and computed brightness temperatures in clear sky conditions in channel 12.05 µm |
| BTDoc(08-12) | 08-12 inter-channel BTDoc difference: BTDoc(08) - BTDoc(12) |
| BTDoc(10-12) | 10-12 inter-channel BTDoc difference: BTDoc(10) - BTDoc(12) |
| $\beta_{eff}12/k$ | Effective microphysical index for the pair of channels 12 and k: $\tau_{a,12}/\tau_{a,k}$ |
| $dT_{k,BB}$ | Systematic error in blackbody brightness temperature in channel k |
| $dT_{k,BG}$ | Systematic error in background brightness temperature in channel k |
| $D_e$ | Effective diameter retrieved by the IIR algorithm |
| $D_e12/k$ | Effective diameter derived from $\beta_{eff}12/k$ |
| $\Delta\varepsilon_{eff}12-k$ | Inter-channel effective emissivity difference: $\varepsilon_{eff,12} - \varepsilon_{eff,k}$ |
| $\Delta T_{BB}$ | Random error in blackbody brightness temperature (all channels) |
| $\Delta T_{BG}$ | Random error in background brightness temperature (all channels) |
| $\Delta Z$ | Geometric thickness |
| $\Delta Z_{eq}$ | IIR equivalent geometric thickness |

| | |
|---|---|
| $\varepsilon_{eff,k}$ | Effective emissivity in IIR channel k |
| $\eta$ | Multiple scattering correction factor |
| IAB | Integrated Attenuated Backscatter at 532 nm |
| IWC | IIR layer equivalent ice water content |
| IWP | Ice water path |
| IWVP | Column-integrated water vapor path |
| k | Used to designate an IIR channel |
| | Channel 08.65 µm: k = 08 |
| | Channel 10.65 µm: k = 10 |
| | Channel 12.05 µm: k = 12 |
| LWC | IIR layer equivalent liquid water content |
| LWP | Liquid water path |
| $N_d$ | Liquid droplets concentration |
| $N_i$ | Ice crystals concentration |
| $R_{k,BB}$ | Blackbody radiance in channel k |
| $R_{k,BG}$ | Background radiance in channel k |
| $R_{k,m}$ | Measured radiance in channel k |
| $T_{base}$ | Temperature at cloud base |
| $T_c$ | Centroid temperature, i.e. thermodynamic temperature at centroid altitude $Z_c$ |
| $T_{k,BB}$ | Blackbody brightness temperature in channel k |
| $T_{k,BG}$ | Background brightness temperature in channel k |
| $T_{k,m}$ | Measured brightness temperature in channel k |
| $T_r(k)$ | Radiative temperature in channel k |
| $T_{top}$ | Temperature at cloud top |
| $\tau_{a,k}$ | Effective absorption optical depth in channel k |
| $\tau_{vis}$ | Visible optical depth |
| $WF_{IIR}$ | IIR weighting function |
| $Z_c$ | Centroid altitude of the 532-nm attenuated backscatter |

**Data availability**

The Version 3 IIR Level 2 track products used in this paper are available at https://doi.org/10.5067/IIR/CALIPSO/L2_Track-Beta-V3-01 (last access: 14 September 2020) and the Version 4 IIR Level 2 track products are available at https://doi.org/10.5067/CALIOP/CALIPSO/CAL_IIR_L2_Track-Standard-V4-20 (last access: 14 September 2020).

The IIR Level 2 track products are also available at the AERIS/ICARE Data and Services Center at https://www.icare.univ-lille.fr/data-access/data-archive-access/?dir=IIR/ (last access 14 September 2020).

## Author contribution

AG and JP defined the changes implemented in the V4 IIR algorithm and wrote the original draft. AG performed the data analysis and prepared the figures. NP was in charge of software development and provided V4 IIR test data. MV provided assistance for the use of the CALIOP data. PD provided the FASRAD and FASDOM radiative transfer models and bulk scattering properties. PY provided the ice habit models from the "TAMUice 2016" database. DM provided the analytical functions derived from in situ measurements. All authors contributed to the review and editing of this paper.

## Competing interests

Author Jacques Pelon is a co-guest editor for the "CALIPSO Version 4 Algorithms and Data Products" special issue in Atmospheric Measurements Techniques but will not participate in any aspects of the editorial review of this manuscript. All other authors declare that they have no conflicts of interest.

## Acknowledgements

The authors are grateful to NASA LaRC, to SSAI (Science Systems and Applications, Inc.), to the Centre National d'Etudes Spatiales (CNES) and to Institut National des Sciences de l'Univers (INSU) for their support. Melody Avery, Bob Holz, and James Campbell are warmly acknowledged for fruitful discussions about the ice LUTs. We thank the AERIS infrastructure for providing access to the CALIPSO products, and for data processing during the development phase. We thank Brian Getzewich and Tim Murray for the processing of the Version 4 IIR Level 2 data at NASA LaRC.

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
