# Peer review of "Version 4 CALIPSO IIR ice and liquid water cloud microphysical properties, Part I: the retrieval algorithms"

_Atmospheric Measurement Techniques, 2020_

## Referee Comment (RC1) · Anonymous Referee #1 · 11 Jan 2021

The paper "Version 4 CALIPSO IIR ice and liquid water cloud microphysical properties, Part I: the retrieval algorithms" discusses the improvements in the V4 algorithms compared to those used in the version 3 (V3) release. The manuscript presents the development of IIR Level 2 data products, the main changes implemented in the V4 IIR Level 2 algorithm, and describes improvements with respect to V3. In addition, the paper provides updates to the scene classification algorithm, describes the changes implemented to compute the effective emissivities in each IIR channel, detailing microphysical properties and retrievals (effective diameter, ice or liquid water path), and estimations on ice crystal and water droplet concentrations. The study falls within the scope of AMT. The authors have done a thorough job and have a rigorous approach.

[Figure]

The manuscript is very well-written and structured, the presentation clear, the language fluent and the quality of the figures high. The results support the conclusions. I recommend publication in AMT, however I recommend the following minor revisions and suggestions before it can proceed to be published.

Comments:

1) Regarding references, in the introduction section, an adequate list of references is provided. However, I would suggest the authors to expand the list of references in order to strengthen the manuscript. For example in the very first paragraph, at the end of line 41 (page 2), and at line 42 (page 2) suitable references could be used. 2) Page 2, line 64: At this point the concept of microphysical index, $\beta$eff., measuring wavelengths, effective absorption optical depths, effective emissivities are introduced in the manuscript. Although the terms are well established, properly explained and presented, this is done later on in the manuscript, leaving a reader to wonder in the early stages of the manuscript. In that case, I would suggest a slight rearrangement, probably would be beneficial for the manuscript, to provide at least brief descriptions at an earlier stage of the manuscript. 3) Page 2, line 69: please provide a more detailed description of the homogeneity criteria used. Although they are detailed in previous studies (Garnier et al., 2012, 2013), as stepping-stone a brief description could be of use. 4) The analysis is mainly in the geographical domain between 60oS and 60oN. Although the biases, the developed algorithms and the improvements are extensively discusses it is not clear the geographical reasons why the analysis is constrained in this domain. I wonder whether the authors can provide an explanation regarding the underlying causes of the geographical preference. 5) Regarding the scene classification, as mentioned, it is based on the characteristics of the layers reported in the CALIOP 5-km cloud and aerosol products. However, as the classification algorithms which is designed to identify suitable scenes containing the required information for the retrievals, sometimes fails to properly classify a cloud/aerosol layer, and moreover in cases of low aerosol/cloud load, due to SNR and CALIOP detection thresholds/capabilities, may

propagate towards the retrievals, the analysis and the uncertainties. It would be beneficial to discuss more extensively in the manuscript the effects of erroneous feature classifications to the retrieval algorithms. 6) Please provide some more information regarding the algorithm performance on thin clouds/cirrus clouds. 7) Is it possible to provide more detailed description on the motivation for changes in V4, through study cases? If the cases are considered to disrupt the flow of the manuscript, I would suggest their inclusion as supplement. 8) The V4 statistics are very interesting, though they may need further explanation in the manuscript. Is it possible to include in the Statistical Table more statistical indicators (e.g. Relative Difference)? 9) In 3.4.2 section, I would suggest to include more information on the correction functions, as mentioned briefly in paragraph 2. 10) 3.4.3. In the Radiative temperature in liquid water clouds, but also in the rest of the section, I would suggest a more detailed approach and description in the manuscript on the uncertainties introduced due to the applied algorithms. If possible, uncertainties should be included in as many presented results and Figures as possible.
* * *

---

## Referee Comment (RC2) · Anonymous Referee #3 · 10 Feb 2021

Comments on AMT manuscript (amt-2020-387) entitled
"Version 4 CALIPSO IIR ice and liquid water cloud microphysical properties,
Part I: the retrieval algorithms" by Anne Garnier, Jacques Pelon, Nicolas Pascal,
Mark A. Vaughan, Philippe Dubuisson, Ping Yang, and David L. Mitchell.

This paper presents retrieval algorithms of cloud micro- and macro-physical properties using Version 4 CALIOP Imaging Infrared Radiometer (IIR) data. It also shows the improvements over Version 3. However, there are several points to improve in the manuscript. The authors must revise their manuscript addressing my following specific comments.

Specific comments

1. On p. 4, lines 124-126: the authors state, "A "Was_Cleared_Flag_1km" SDS is now available in the V4 IIR product, which reports the number of CALIOP single-shot clouds in the atmospheric column seen by the 1-km IIR pixel that were cleared from the 5-km layer products." Spell out 'SDS'.

2. On p. 5, Fig. 1b: at $\varepsilon_{eff,12} < {\sim}0.4$, $dT_{k,BG} = 0.1$ K (red dashed line) deviates more from 0 than $dT_{k,BG} = 1$ K (black dashed line). This is opposite to what I expect. Why?

3. On p. 6, lines 211-212: the authors state, "Underestimating $T_r$ (and therefore TOA $T_{BB}$) yields under-estimates in $\varepsilon_{eff,12}$ and the microphysical indices." What is difference between 'radiative temperature $T_r$' and 'TOA $T_{BB}$'?

4. On p. 7, lines 246-247: the authors state, "In contrast, the V3 median 10-12 and 08-12 inter-channels biases were up to - 0.7 K and – 1.8 K, respectively, at IWVP = 5 g.m$^{-2}$."  '5 g.m$^{-2}$' should be '5 g.cm$^{-2}$'.

5. On p. 8, lines 270-271: the authors state, "In V4, the mean absolute inter-channel differences are smaller than 0.1 K globally.". What is the difference between 'mean absolute inter-channel difference' and 'mean absolute deviation (MAD) of the differences between observed and computed brightness temperatures' in Table 2?

6. Fig. 3(j) and Fig. 4(a): In summer, peak of V4 daytime (red) is more deviates more from 0 than peak of V3 daytime (red). Why?

7. On p. 14, lines 379-380: the authors state, "For daytime data, both $T_r$ and $T_m$ are lower in the apparent cloud than at night, and even below ($T_m > T_{base}$), which is at least in part due to the smaller daytime apparent thickness.". However, for daytime data, both $T_r$ and $T_m$ are higher than at night in Fig. 7(c). How do you reconcile these opposite facts?

8. On p. 16, lines 431 and 435: '$\beta_{12/k}$' should be '$\beta_{\text{eff}12/k}$'.

9. On p. 19, lines 493-494: the authors state, "For a given De, $\beta_{\text{eff}12/10}$ is notably larger when $N(D)_1$ is not modified (blue and red solid lines) than when $N(D)_1$ is forced to zero (blue and red dashed lines), because the presence of small particles in the unmodified PSD increases $\beta_{\text{eff}12/10}$ faster than $D_e$.". 'faster' should be rephrased.

10. On p. 22, Eq. (12): Define the IIR weighting function $WF_{\text{IIR}}(z)$ used in Eq. (12).

11. On p. 25, Eq. (A4): Define $\varepsilon_{12,x}$ and $\varepsilon_{k,x}$ used in Eq. (A4).

Technical corrections

1. On p. 8, line 252: the authors state, "Over-plotted in green in the median MERRA-2 surface temperature; (b): number of IIR pixels.". 'in the median' should be 'is the median'.

2. On p. 28, line 748: the authors state, "in the 10-mm window region". '10-mm' should be '10-$\mu$m'.

---

## Referee Comment (RC3) · Bryan A. Baum (Referee) · 17 Feb 2021

This is a very thorough article that discusses improvements and modifications to the cloud microphysical and optical property retrieval algorithms in Version 4 of the CALIPSO products from the Imaging Infrared Radiometer (IIR).

This paper is suitable for publication but could use some improvements on general readability. For me, the issue is that the manuscript is full of terms for both measured and simulated radiances and brightness temperatures, emissivities, beta indices for the microphysical retrievals, and also a large number of acronyms and abbreviations. This became quite difficult to track since many these parameters have multiple subscripts -
it is very difficult to keep these straight. A glossary of parameters/subscripts/acronyms would have really helped me.

Line 20: the authors discuss reducing biases found at very small emissivities in V3 of their products, both here and in Section 3.2.1 beginning at line 190. My interpretation of this is that there is a significant low biases in the ice cloud microphysical indices at very low values of the cloud emissivity, which is the same thing as stating that there is a bias at very low ice cloud optical depths. On lines 26/27, the authors state that V4 improved retrievals in ice clouds having large optical depths. My point is to be consistent in the use of cloud emissivity or cloud optical depth. In fact, lines 569-570 say this very clearly: "The IIR Level 2 algorithm has been modified in the V4 data release to improve the accuracy of the microphysical indices in clouds of very small (close to 0) and very large (close to 1) effective emissivities." Perhaps this sentence should also be in the Abstract.

Line 25: why is the IIR channel at 8.65 microns written as 08.65 here and throughout the manuscript? Is there a reason for including a leading zero on this wavelength?

Line 26: suggest changing "aimed at improving" to "improved"

Line 31: define what is meant by "dense ice clouds" here and on lines 112, 122, 296, and 587.

Line 33: mostly a comment: this is the first of 24 references to "ice crystal models" or something similar in the text. The term "crystal" generally suggests a pristine shape such as a column or plate. The term "particle" includes all habits, pristine or very complex. Naturally occurring ice particles mostly defy description. This article more properly describes the adoption of two "ice habit models" composed of either single hexagonal columns (first found on line 33) or aggregates of columns (line 34).

Line 41: add a sentence to provide background and a reference for the A-Train for those readers who may not be familiar with it.

[Figure]

Line 41: define spectrum of wavelengths meant by visible and infrared

Line 42: suggest changing "combination of infrared" to "combination of passive infrared"

Lines 127-128: provide a description of the new types of scenes that have been introduced when at least one cleared cloud is present in the column

Lines 185 and 186: define what is meant specifically by optically very thin and very thick cloud here.

Line 218: suggest changing "Earth Surface" to "surface"

Line 218: interpolated atmospheric profiles: how many layers/levels are in the interpolated profiles? Do these profiles include trace gases? Are these augmented at very high altitudes by a climatology, e.g., for ozone if this is part of the profile?

Line 222: suggest changing "thanks to the advances" to "to take advantage of recent advances"

Section 3.4.1, lines 313-317: the discussion on calculating a centroid altitude and temperature for multi-layered cloud cases is a bit confusing to me. If a vertical column contains optically thin ice cloud overlying a low-level water cloud, can the resulting centroid be in the mid-troposphere where there is no cloud layer? If this is a possibility, there should be a flag provided to indicate that multilayered clouds are present for that retrieval so that these cases can be filtered out if so desired. More specifically, the flag should be provided with the cloud properties such as the centroid altitude and temperature so that a user does not have to look at potentially multiple products (e.g., cloud mask or cloud phase) to find this detail. The availability of a flag would certainly be of help when comparing your cloud product to a simulated cloud field based on, for example, large eddy simulations.

In Section 4 somewhere, it would be quite interesting to know the range of the effective diameter (De) values for ice clouds inferred from both V3 and V4. Does the range

change between V3 and V4? Additionally, does the range ever approach the boundaries of the LUT, either very low or very high values? How often does this happen?

---

## Author Comment (AC1) · 17 Mar 2021

Response to anonymous referee #1

The authors are thankful to the referee for his/her thorough review of the paper.

Our responses are detailed below. The manuscript has been revised accordingly, and is clearly improved.

In the following, the reviewer's comments are in black, and our answer to each comment is in red.

Comments:
1) Regarding references, in the introduction section, an adequate list of references is provided. However, I would suggest the authors to expand the list of references in order to strengthen the manuscript. For example in the very first paragraph, at the end of line 41 (page 2), and at line 42 (page 2) suitable references could be used.
We added the following references at the beginning of the introduction:

Bodas-Salcedo, A., Hill, P. G., Furtado, K., Williams, K. D., Field, P. R., Manners, J. C., Hyder, P. and Kato, S.: Large contribution of supercooled liquid clouds to the solar radiation budget of the Southern Ocean, J. Climate, **29**, 4213-4228, https://doi.org/10.1175/JCLI-D-15-0564.1, 2016.

Muhlbauer, A., McCoy, I. L., and Wood, R.: Climatology of stratocumulus cloud morphologies: microphysical properties and radiative effects, Atmos. Chem. Phys., 14, 6695–6716, https://doi.org/10.5194/acp-14-6695-2014, 2014.

Stephens et al. (2002).

Stephens, G., Winker, D., Pelon, J., Trepte, C., Vane, D., Yuhas, C., L'Ecuyer, T., and Lebsock, M.: CloudSat and CALIPSO within the A-Train: Ten years of actively observing the Earth system, Bull. Amer. Meteor. Soc., 99(3), 569–581, https://doi.org/10.1175/BAMS-D-16-0324.1, 2018.

Duncan, D. I. and Eriksson, P.: An update on global atmospheric ice estimates from satellite observations and reanalyses, Atmos. Chem. Phys., 18, 11205–11219, https://doi.org/10.5194/acp-18-11205-2018, 2018.

Stubenrauch, C. J., Caria, G., Protopapadaki, S. E., and Hemmer, F.: 3D radiative heating of tropical upper tropospheric cloud systems derived from synergistic A-Train observations and machine learning, Atmos. Chem. Phys., 21, 1015–1034, https://doi.org/10.5194/acp-21-1015-2021, 2021.

2) Page 2, line 64: At this point the concept of microphysical index, $\beta\_eff$., measuring wavelengths, effective absorption optical depths, effective emissivities are introduced in the manuscript. Although the terms are well established, properly explained and presented, this is done later on in the manuscript, leaving a reader to wonder in the early stages of the manuscript. In that case, I would suggest a slight rearrangement, probably would be beneficial for the manuscript, to provide at least brief descriptions at an earlier stage of the manuscript.

We followed the reviewer's suggestion and added a brief description of the emissivity retrievals in this paragraph of the introduction. The text now reads (changes in italic):
"Effective emissivities and microphysical retrievals are reported in the IIR Level 2 data products. The Version 3 (V3) products released in 2011 used the V3 CALIOP data products. As described in G12 and G13, they were focused on retrievals of ice cloud properties. *Effective emissivity represents the fraction of the upward radiation absorbed and re-emitted by the cloud system. The IIR 1-km pixel is assumed to be fully cloudy and the qualifying adjective "effective" refers here to the contribution from scattering. The retrievals are applied to suitable scenes that are identified and characterized by taking advantage of co-located CALIOP retrievals. Effective emissivity in each IIR channel is retrieved after determining the background radiance that would be observed in the absence of the studied cloud system and the blackbody radiance that would be observed if the cloud system were a blackbody source*. Unlike the well-known split-window technique (Inoue, 1985), which relies on the analysis of inter-channel brightness temperature differences, IIR microphysical retrievals use the concept of microphysical index ($\beta_{eff}$) proposed by Parol et al. (1991). This concept is applied to the pairs of IIR channels at 12.05 μm and 10.6 μm and at 12.05 μm and 08.65 μm, with $\beta_{eff}$12/10 and $\beta_{eff}$12/08 defined as, respectively, the 12.05-to-10.6 ratio and the 12.05-to-08.65 ratio of the effective absorption optical depths. The latter are derived from *the* cloud effective emissivities retrieved in each of the three channels. The microphysical indices are interpreted in terms of $D_e$ by using look-up tables (LUTs) built for several ice *habit* models. $D_e$ is retrieved using the ice *habit* model that provides the best agreement with the observations in terms of relationship between $\beta_{eff}$12/10 and $\beta_{eff}$12/08. Total water path is then estimated using IIR $D_e$ *and visible optical depth estimated from IIR* effective emissivities."

The sentence: "*The IIR 1-km pixel is assumed to be fully cloudy and the qualifying adjective "effective" refers here to the contribution from scattering*", was moved from Sect. 3.1 to the introduction.

3) Page 2, line 69: please provide a more detailed description of the homogeneity criteria used. Although they are detailed in previous studies (Garnier et al., 2012, 2013), as stepping-stone a brief description could be of use.

Thank you for this suggestion. A brief description has been added and the text now reads (changes in italic):
"Retrievals along the CALIOP track are extended to the IIR swath *by assigning to each swath pixel the retrievals in the radiatively most similar track pixel at a maximum distance of 50 km (G12). This most similar track pixel is found by minimizing the mean absolute difference between the brightness temperatures in the three channels, with an upper threshold set to 1 K. Retrievals along the CALIOP track and over the IIR swath* are reported in the IIR Level 2 track and swath data products, respectively."

4) The analysis is mainly in the geographical domain between 60oS and 60oN. Although the biases, the developed algorithms and the improvements are extensively discussed it is not clear the geographical reasons why the analysis is constrained in this domain. I wonder whether the authors can provide an explanation regarding the underlying causes of the geographical preference.

We should have explained that we chose this geographical domain to ensure that the dataset is not contaminated by sea ice (retrievals over surface types other than oceans will be presented in an upcoming paper). This is now explained in Sect. 3.3.1 where Fig. 2 is described (new text in italic):

"The results are shown for 6 months of nighttime data in 2006 (from July through December) between 60° S and 60° N *to ensure that the dataset is not contaminated by sea ice*."

5) Regarding the scene classification, as mentioned, it is based on the characteristics of the layers reported in the CALIOP 5-km cloud and aerosol products. However, as the classification algorithms which is designed to identify suitable scenes containing the required information for the retrievals, sometimes fails to properly classify a cloud/aerosol layer, and moreover in cases of low aerosol/cloud load, due to SNR and

CALIOP detection thresholds/capabilities, may propagate towards the retrievals, the analysis and the uncertainties. It would be beneficial to discuss more extensively in the manuscript the effects of erroneous feature classifications to the retrieval algorithms.

Thank you for this question.

a) The IIR operational algorithm is applied regardless of the confidence in the CALIOP cloud/aerosol classifications. However, we are now explaining in the text that this confidence in the classifications is reported in the product. Cloud/aerosol mis-classifications are expected to be associated to no or low confidence in the feature type classification, and therefore can be easily filtered out using the Quality Assessment flags reported in the IIR product. The impact of a layer mis-classification depends on the contribution of this layer to the measured infrared radiances. In a simple case where the column includes one aerosol layer mis-classified as a cloud, the microphysical retrievals will likely fail because the LUTs are designed for cloud retrievals. In another simple case where the column includes one cloud layer mis-classified as an aerosol, no microphysical retrievals will be provided.

Analyses of both CALIOP and IIR retrievals in case of possible cloud/aerosol mis-classifications are required before we can fully address this question in a satisfactory manner. This would require first to establish why the CALIOP algorithm had no or low confidence in the classifications and then to examine the IIR retrievals to assess whether there is evidence of mis-classification. Such studies could help better characterize the performance of our combined retrievals for these challenging cases.

b) CALIOP detection capabilities are typically superior to the sensitivity required for the IIR algorithm. Thus, the scene classification is based on layers detected by the CALIOP algorithm at 5-km and 20-km horizontal averaging intervals (Vaughan et al., 2009), while layers detected by CALIOP at 80-km horizontal averaging intervals are ignored because they are optically too thin to be seen by IIR.

Nevertheless, CALIOP detection capabilities are limited when a layer fully attenuates the CALIOP signal, so that the lower part of the cloud is missed by CALIOP. For these opaque clouds, the true base is a priori not detected, which introduces uncertainties in the determination of the radiative temperature in ice clouds as discussed in Sect. 3.4.2.

c) Changes in the text:

The text at the beginning of Sect. 2 now reads (changes in italic):
"Both in V4 and in V3, the first task of the IIR algorithm is to classify the pixels in the scenes being viewed. This scene classification is based on the characteristics of the layers reported in the CALIOP 5-km cloud and aerosol products *for layers detected by the CALIOP algorithm at 5-km and 20-km horizontal averaging intervals (Vaughan et al., 2009). This classification* is designed to identify suitable scenes containing the required information for effective emissivity retrievals."

New reference:
Vaughan, M., Powell, K., Kuehn, R., Young, S., Winker, D., Hostetler, C., Hunt, W., Liu, Z., McGill, M., and Getzewich, B.: Fully automated detection of cloud and aerosol layers in the CALIPSO lidar measurements, J. Atmos. Oceanic Technol., 26, 2034–2050, https://doi.org/10.1175/2009JTECHA1228.1, 2009.

And the following text has been added at the end of Sect. 2:
*"A lot of other parameters characterizing the scenes are reported in the V4 IIR product. Among them are the number of layers in the cloud system, as well as an Ice Water Flag which informs the user about the phase of the cloud layers included in the system, as assigned by the V4 CALIOP Ice/Water phase algorithm (Avery et al., 2020). A companion Quality Assessment Flag reports the mean confidence in the feature type*

*(i.e., cloud or aerosol) classification (Liu et al., 2019) and in the phase assignment for these cloud layers. The product also includes the number of tropospheric dust layers and of stratospheric aerosols layers in the column and the mean confidence in the feature type classification. All the suitable scenes are processed regardless of the confidence in the classifications and phase assignments reported in the CALIOP products, so that the user can define customized filtering criteria adapted to specific research objectives."*

New reference:

Liu, Z., Kar, J., Zeng, S., Tackett, J., Vaughan, M., Avery, M., Pelon, J., Getzewich, B., Lee, K.-P., Magill, B., Omar, A., Lucker, P., Trepte, C., and Winker, D.: Discriminating between clouds and aerosols in the CALIOP version 4.1 data products, Atmos. Meas. Tech., 12, 703-734, https://doi.org/10.5194/amt-12-703-2019, 2019.

6) Please provide some more information regarding the algorithm performance on thin clouds/cirrus clouds. In the description of Fig.1, we added in the text that $\varepsilon_{eff,12} = 0.1$ corresponds to optical depth $\sim 0.2$, and to a thin cirrus cloud. The text reads (changes in italic):

"at $\varepsilon_{eff,12} \sim 0.1$ *(or optical depth $\sim 0.2$, corresponding to a thin cirrus cloud)*, $\beta_{eff}12/10$ (dashed line) is decreased…"

7) Is it possible to provide more detailed description on the motivation for changes in V4, through study cases? If the cases are considered to disrupt the flow of the manuscript, I would suggest their inclusion as supplement.

The changes in V4 are motivated by the need to reduce systematic biases in the IIR V3 products that were made evident after statistical analyses of the retrievals. By accumulating a sufficient number of individual retrievals, typically over a month, the random noise could be significantly reduced, but the systematic biases remained. These biases are not unambiguously detected through case studies, because they can be hidden by the noise. We do not think that showing case studies would provide useful additional information.

8) The V4 statistics are very interesting, though they may need further explanation in the manuscript. Is it possible to include in the Statistical Table more statistical indicators (e.g. Relative Difference)?

In this section, we examine the differences between the observed and computed brightness temperatures in order to assess the errors in the computed background radiances used in the effective emissivity retrievals. The relevant indicator is the error in $T_{k,BG}$, rather than the relative error in $T_{k,BG}$, which is why relative differences are not provided in Table 2.

We tried to clarify the text by adding the following sentence at the beginning of Sect. 3.3.2:

*"In order to assess the errors in the computed background radiances used in the effective emissivity retrievals ($R_{k,BG}$, see Eq. 1) and in the corresponding computed brightness temperatures ($T_{k,BG}$), we analyzed distributions of BTDoc for different latitudes and seasons."*

Furthermore, we modified the text at the end of this section, which now reads (changes in italic):

"*Thus, the analysis of these inter-channel distributions shows that* the uncertainty in computed $T_{k,BG}$ *can be* taken identical in all channels. *Based on the standard deviations in BTDoc(12), the random error $\Delta T_{BG}$ is set to the conservative value $\pm 1$ K for all channels*."

9) In 3.4.2 section, I would suggest to include more information on the correction functions, as mentioned briefly in paragraph 2.

We now include the two important equations established in G15, and re-organized the beginning of the section, which now reads (changes in italic):

*"As demonstrated in G15, the radiative temperature $T_r(k)$ in channel k is the brightness temperature associated with the centroid radiance of the attenuated infrared emissivity profile within the cloud. For a cloud containing a number, n, of vertical bins, i, of resolution dz, with i = 1 to i = n from base to top, this centroid radiance can be written as a function of radiance $R_k(i)$ of bin i and CALIOP particulate (i.e., cloud) extinction coefficient, $\alpha_{part}(i)$, as:*

$$R_k = \frac{\sum_{i=1}^{i=n}\left(1 - e^{-\left[\alpha_{part}(i)\cdot\delta z\middle/ r\right]}\right)\cdot R_k(i).e^{-\sum_{j=i+1}^{j=n+1}\left[\alpha_{part}(j)\cdot\delta z\middle/ r\right]}}{\varepsilon_{eff,k}} \tag{5}$$

*The term $\alpha_{part}(i).\delta z/r$ in Eq. (5) is the absorption optical depth in bin i. The ratio, r, of CALIOP optical depth to IIR absorption optical depth is taken equal to 2 (G15). The radiance $R_k(i)$ is determined from the thermodynamic temperature in bin i.*

*On the other hand, $T_c$ is the temperature at the centroid altitude of the attenuated 532 nm backscatter coefficient profile, which is written as a function of altitude Z(i) at bin i and $\alpha_{part}(i)$ as:*

$$Z_c = \frac{\sum_{i=1}^{i=n}Z(i)\cdot\left(\beta_{part}(i) + \beta_{mol}(i)\right)\cdot e^{-2\sum_{j=i}^{j=n}\left[\eta\alpha_{part}(j)+\alpha_{mol}(j)\right]\cdot\delta z}}{\sum_{i=1}^{i=n}\left(\beta_{part}(i) + \beta_{mol}(i)\right)\cdot e^{-2\sum_{j=i}^{j=n}\left[\eta\alpha_{part}(j)+\alpha_{mol}(j)\right]\cdot\delta z}} \tag{6}$$

*In Eq. (6), $\beta_{part}(i)$ is the cloud particulate backscatter in bin i, $\alpha_{mol}(i)$ and $\beta_{mol}(i)$ are the molecular extinction coefficient and backscatter, respectively, and $\eta$ is the ice cloud multiple scattering correction factor (Young et al. (2018) and references therein).*

Using V3 CALIOP extinction and *backscatter* profiles in semi-transparent ice clouds, the $T_r$-$T_c$ difference was found to increase with both cloud optical depth and geometric thickness (G15).

Because the CALIOP extinction profiles are not used in the IIR operational algorithm, the approach in V4 was to establish parameterized correction functions, $T_r(k)$ - $T_c$, for each channel k, and to correct the initial estimate $T_c$ that was used in V3 as $T_r(k) = T_c + [T_r(k) - T_c]$. These correction functions were derived off-line from the statistical analysis of a series of simulated extinction and attenuated backscatter profiles. In order to reproduce the variability associated to the various possible shapes of the extinction profiles, we chose to use actual V4 CALIOP profiles (8,000 profiles were used) rather than synthetic profiles. These initial CALIOP profiles were derived from single-layered semi-transparent clouds classified with high confidence as randomly oriented ice (ROI) by the V4 ice/water phase algorithm (Avery et al., 2020). Each CALIOP extinction (and backscatter) profile was scaled to simulate several pre-defined optical depths *corresponding to several pre-defined effective emissivities using r = 2*, and the attenuated backscatter profile was simulated by applying the required attenuation to the simulated total (molecular and particulate) backscatter profile. The simulations *of $T_r(k)$ using Eq. (5) and of $T_c$ using Eq. (6)* were carried out for $\varepsilon_{eff,k}$ ranging between 0.1 (or $\tau_{a,k}$ = 0.1, see Eq. (2)) and 0.99 (or $\tau_{a,k}$ = 4.6). Variations of $T_r(k)$ - $T_c$ with $\eta$ between 0.5 and 0.8 were also analyzed in order to cover the range of temperature-dependent values used in V4 (G15; Young et al., 2018). Variations with $\eta$ were not discussed in G15 because $\eta$ was taken constant and equal to 0.6 in V3.

The $T_r(k)$ – $T_c$ differences were examined against the "thermal thickness" of the clouds; that is, the difference between the temperatures at cloud base ($T_{base}$) and at cloud top ($T_{top}$). *Ninety percent of the CALIOP profiles used for this analysis had $T_{base}$ – $T_{top}$ between 10 and 50 K.*"

10) 3.4.3. In the Radiative temperature in liquid water clouds, but also in the rest of the section, I would suggest a more detailed approach and description in the manuscript on the uncertainties introduced due to the applied algorithms. If possible, uncertainties should be included in as many presented results and Figures as possible.

In this section, we present the radiative temperature in ice clouds (3.4.2) and in liquid water clouds.

Radiative temperature in ice clouds (3.4.2)
The correction functions presented in this section were obtained from median values of $(T_r – T_c)/(T_{base} – T_{top})$ as a function of $T_{base}$ – $T_{top}$ based on 8,000 CALIOP profiles. To evaluate the error in the corrections, we added comparisons of the radiative temperatures derived directly using the CALOP extinction profiles to the radiative temperatures derived from the algorithm. The following text and a new Table are now added in Sect. 3.4.2 after Fig. 6:
*"The errors in the ice cloud radiative temperature corrections were assessed by comparing $T_r$ derived directly using the CALIOP extinction profiles with $T_r$ derived from Eq. (7). The statistics obtained from the same 8,000 CALIOP profiles as above are provided in Table 3, for both the $T_r$ – $T_c$ correction and the correction error, for channel 12.05 μm. These statistics are provided for $\varepsilon_{eff,12}$ equal to 0.2, 0.6, and 0.99, and using η equal to the extreme values 0.5 and 0.8. The median and mean correction errors are smaller than 0.25 K and significantly smaller than the median and mean corrections, which are found between 0.8 K and 5 K. The standard deviations of the correction errors are between 0.66 and 1.2 K at η = 0.5 and between 0.7 and 1.75 K at η = 0.8, while their mean absolute deviations are smaller than 1.25 K. These quantities represent the estimated random error in the cloud radiative temperature correction resulting from the variability in the shape of the extinction profiles.*

*Table 3: Statistics (median, mean, standard deviation (STD), and mean absolute deviation (MAD)) of the $T_r$-$T_c$ correction at 12.05 μm and of correction errors for $\varepsilon_{eff,12}$ equal to 0.2, 0.6, and 0.99, using η equal to 0.5 and 0.8. Channel 12.05 μm.*

| | | $T_r$ -$T_c$ correction (K) | | | Correction error (K) | | |
|---|---|---|---|---|---|---|---|
| | | $\varepsilon_{eff,12} = 0.2$ | $\varepsilon_{eff,12} = 0.6$ | $\varepsilon_{eff,12} = 0.99$ | $\varepsilon_{eff,12} = 0.2$ | $\varepsilon_{eff,12} = 0.6$ | $\varepsilon_{eff,12} = 0.99$ |
| η = 0.5 | Median | 0.81 | 2.04 | 3.07 | 0.08 | -0.01 | 0.01 |
| | Mean | 0.92 | 2.22 | 3.43 | 0.1 | 0.02 | 0.13 |
| | STD | 0.55 | 1.16 | 1.98 | 0.66 | 0.75 | 1.2 |
| | MAD | 0.43 | 0.93 | 1.56 | 0.46 | 0.53 | 0.84 |
| η = 0.8 | Median | 1.32 | 3.8 | 4.50 | 0.17 | -0.01 | 0.01 |
| | Mean | 1.45 | 4.11 | 5.01 | 0.23 | 0.00 | 0.17 |
| | STD | 0.78 | 2.08 | 2.86 | 0.7 | 0.98 | 1.74 |
| | MAD | 0.62 | 1.65 | 2.26 | 0.50 | 0.70 | 1.24 |

The error in the radiative temperature estimates is difficult to assess unambiguously, because there is no definite reference to compare the observations with. The errors can be due to the algorithm or to the MERRA-2 temperature profiles.

For ice clouds (Sect. 3.4.2), our main concern is for opaque clouds, because CALIOP sees an apparent base and not the true base. We compared $T_r$ with $T_{base}$ and $T_{top}$ in opaque clouds (Fig. 7), and found that the V4 results are on average in excellent agreement with recent analyses by Stubenrauch et al. (2017), who retrieve a radiative height from AIRS infrared observations. We also compared the measured brightness temperature and the radiative temperature and show that the maximum possible bias is equal to 1.5 K on average.

For water clouds (Sect. 3.4.3), we compared directly the measurements in opaque water clouds with the TOA blackbody temperatures derived from $T_r$ and the radiative transfer model, and we provide with statistics. We also illustrate the impact of the temperature profiles used for the retrievals.

---

## Author Comment (AC2) · 17 Mar 2021

Response to anonymous referee #3

The authors are thankful to the referee for his/her thorough review of the paper.

Our responses are detailed below, and the manuscript has been revised accordingly. In the following, the reviewer's comments are in black, and our answer to each comment is in red.

Specific comments
1. On p. 4, lines 124-126: the authors state, "A "Was_Cleared_Flag_1km" SDS is now available in the V4 IIR product, which reports the number of CALIOP single-shot clouds in the atmospheric column seen by the 1-km IIR pixel that were cleared from the 5-km layer products."
Spell out 'SDS'.
For simplicity, we replaced 'SDS' with 'parameter'.

2. On p. 5, Fig. 1b: at $\varepsilon_{eff,12} < \sim 0.4$, $dT_{k,BG} = 0.1$ K (red dashed line) deviates more from 0 than $dT_{k,BG} = 1$ K (black dashed line). This is opposite to what I expect. Why?
The red dashed line corresponds to $dT_{12,BG} = 0$ K and $dT_{k,BG} = 0.1$ K. Channel k is biased but not channel 12, and as a result, the inter-channel 12-k effective emissivity difference is biased.

The black dashed line corresponds to $dT_{12,BG} = 1$ K and $dT_{k,BG} = 1$ K. Both channels are biased by the same quantity in terms of brightness temperature, but this induces anyway a bias in the inter-channel 12-k effective emissivity difference, but which differs from the other bias shown in red.

3. On p. 6, lines 211-212: the authors state, "Underestimating Tr (and therefore TOA TBB) yields under-estimates in $\varepsilon_{eff,12}$ and the microphysical indices." What is difference between 'radiative temperature Tr' and 'TOA TBB'?
"TOA $T_{BB}$' is the Top Of Atmosphere blackbody brightness temperature corresponding to the TOA blackbody radiance determined from $T_r$ and the FASRAD model. These quantities are defined in Sect. 3.1. The difference between $T_{BB}$ and $T_r$ depends on the atmospheric absorption above the cloud.

4. On p. 7, lines 246-247: the authors state, "In contrast, the V3 median 10-12 and 08-12 interchannels biases were up to - 0.7 K and –1.8 K, respectively, at IWVP = 5 g.m-2.". '5 g.m-2'
should be '5 g.cm-2'.
Fixed.

5. On p. 8, lines 270-271: the authors state, "In V4, the mean absolute inter-channel differences are smaller than 0.1 K globally.". What is the difference between 'mean absolute inter-channel difference' and 'mean absolute deviation (MAD) of the differences between observed and computed brightness temperatures' in Table 2?
Indeed, this is confusing.
We replaced "the mean absolute inter-channel differences are smaller than 0.1 K globally"
with
"*the absolute values of* the mean inter-channel differences are smaller than 0.1 K globally"

Fig. 3(j) and Fig. 4(a): In summer, peak of V4 daytime (red) is more deviates more from 0 than peak of V3 daytime (red). Why?

BTDoc(12) is overall less latitude-dependent in V4 than in V3, owing to the reduced bias related to IWVP in V4. In these two cases, it seems that the biases related to IWVP in V3 and those related to sea surface temperature are of opposite signs and such that V3 deviates less than V4. Note that the V3 distributions are nevertheless larger than the V4 distributions, suggesting larger biases related to IWVP in V3.

7. On p. 14, lines 379-380: the authors state, "For daytime data, both Tr and Tm are lower in the apparent cloud than at night, and even below (Tm > Tbase), which is at least in part due to the smaller daytime apparent thickness.". However, for daytime data, both Tr and Tm are higher than at night in Fig. 7(c). How do you reconcile these opposite facts?
We checked both the text and Fig. 7c, and we think that there is no mistake.
In Fig. 7c, both $T_r$ and $T_m$ have $(T-T_{top})/(T_{base}-T_{top})$ larger for daytime data than at night, which means that both $T_r$ and $T_m$ are closer to the base for daytime data than at night, and therefore that both $T_r$ and $T_m$ are lower in altitude for daytime data than at night.

8. On p. 16, lines 431 and 435: '$\beta_{12/k}$' should be '$\beta_{eff12/k}$'.
Fixed

9. On p. 19, lines 493-494: the authors state, "For a given De, $\beta_{eff12/10}$ is notably larger when N(D)1 is not modified (blue and red solid lines) than when N(D)1 is forced to zero (blue and red dashed lines), because the presence of small particles in the unmodified PSD increases $\beta_{eff12/10}$ faster than De.". 'faster' should be rephrased.
"Faster" has been replaced with "more rapidly".

10. On p. 22, Eq. (12): Define the IIR weighting function WFIIR(z) used in Eq. (12).
We added a new equation to define the IIR weighting function and re-organized the beginning of this section as follows (changes in italic). In the new text, we refer to a new Eq. (5) which has been added in Sect. 3.4.2 after comments by referee #1. The notations not specified here are introduced with Eq. (5):

"The IIR retrievals are all tied to the retrieved effective emissivities. As demonstrated in G15, $\varepsilon_{eff,k}$ is the vertical integration of an attenuated effective emissivity profile, which can be determined from the CALIOP extinction profile, $\alpha_{part}(z)$. *Looking at Eq. (5) used to derive the cloud radiative temperature and ultimately establish the correction functions presented in Sect. 3.4.2, we see that we can define an IIR weighting function $WF_{IIR}(i)$ as*:

$$WF_{IIR}(i) = \frac{1-e^{-\left[\alpha_{part}(i)\cdot\delta z / r\right]}}{\varepsilon_{eff,k}} \cdot e^{-\sum_{j=i+1}^{j=n+1}\left[\alpha_{part}(j)\cdot\delta z / r\right]} \tag{14}$$

*This applies to semi-transparent clouds whose true base is detected by CALIOP. This concept has been used in M18 to compute an equivalent effective thickness seen by IIR, ΔZeq, derived from the geometric thickness, ΔZ, as:*

$$\frac{1}{\Delta Zeq} = \frac{1}{\Delta Z} \times \frac{1}{\tau_{vis}} \times \sum_{i=1}^{i=n}\alpha_{part}(i).WF_{IIR}(i).\delta z \tag{15}$$

11. On p. 25, Eq. (A4): Define $\varepsilon_{12,x}$ and $\varepsilon_{k,x}$ used in Eq. (A4).
This was a mistake.

The term $\varepsilon_{12,x}$ should be $\varepsilon_{eff,12}$ and likewise $\varepsilon_{k,x}$ should be $\varepsilon_{eff,k}$
The corrected equation is:

$$\frac{\left(d\beta_{eff}12/k\right)_x}{\beta_{eff}12/k} = \frac{-d\varepsilon_{12,x}}{\left(1-\varepsilon_{eff,12}\right)\ln(1-\varepsilon_{eff,12})} + \frac{d\varepsilon_{k,x}}{\left(1-\varepsilon_{eff,k}\right)\ln(1-\varepsilon_{eff,k})} \tag{A4}$$

Technical corrections
1. On p. 8, line 252: the authors state, "Over-plotted in green in the median MERRA-2 surface temperature; (b): number of IIR pixels.". 'in the median' should be 'is the median'.
Fixed

2. On p. 28, line 748: the authors state, "in the 10-mm window region". '10-mm' should be '10-μm'.
Fixed

---

## Author Comment (AC3) · 17 Mar 2021

Response to Bryan A. Baum

The authors are thankful to the referee for his thorough review of the paper.

Our responses are detailed below. The manuscript has been revised accordingly, and is clearly improved.

In the following, the reviewer's comments are in black, and our answer to each comment is in red.

A glossary of parameters/subscripts/acronyms would have really helped me.
We recognize that a lot parameters with many subscripts are used in this paper. We followed the reviewer suggestion and included a new Appendix C with a glossary for the most important parameters used in the manuscript. Appendix C will read:

**Appendix C: Glossary**

| Notation | Description |
|---|---|
| $\alpha_{abs,eq}(k)$ | IIR equivalent absorption coefficient in channel k |
| $\alpha_{part}$ | CALIOP particulate extinction coefficient |
| BTDoc | Difference between observed and computed brightness temperatures in clear sky conditions, channel not specified |
| BTDoc(12) | Difference between observed and computed brightness temperatures in clear sky conditions in channel 12.05 µm |
| BTDoc(08-12) | 08-12 inter-channel BTDoc difference: BTDoc(08) - BTDoc(12) |
| BTDoc(10-12) | 10-12 inter-channel BTDoc difference: BTDoc(10) - BTDoc(12) |
| $\beta_{eff}12/k$ | Effective microphysical index for the pair of channels 12 and k: $\tau_{a,12}/\tau_{a,k}$ |
| $dT_{k,BB}$ | Systematic error in blackbody brightness temperature in channel k |
| $dT_{k,BG}$ | Systematic error in background brightness temperature in channel k |
| $D_e$ | Effective diameter retrieved by the IIR algorithm |
| $D_e12/k$ | Effective diameter derived from $\beta_{eff}12/k$ |
| $\Delta\varepsilon_{eff}12-k$ | Inter-channel effective emissivity difference: $\varepsilon_{eff,12} - \varepsilon_{eff,k}$ |
| $\Delta T_{BB}$ | Random error in $T_{k,BB}$ (all channels) |
| $\Delta T_{BG}$ | Random error in background brightness temperature (all channels) |
| $\Delta Z$ | Geometric thickness |
| $\Delta Z_{eq}$ | IIR equivalent geometric thickness |
| $\varepsilon_{eff,k}$ | Effective emissivity in IIR channel k |
| $\eta$ | Multiple scattering correction factor |
| IAB | Integrated Attenuated Backscatter at 532 nm |
| IWC | IIR layer equivalent ice water content |
| IWP | Ice water path |
| IWVP | Column integrated water vapor path |
| k | Used to designate an IIR channel
Channel 08.65 µm: k = 08
Channel 10.65 µm: k = 10
Channel 12.05 µm: k = 12 |
| LWC | IIR layer equivalent liquid water content |
| LWP | Liquid water path |

| $N_d$ | Liquid droplets concentration |
|---|---|
| $N_i$ | Ice crystals concentration |
| $R_{k,BB}$ | Blackbody radiance in channel k |
| $R_{k,BG}$ | Background radiance in channel k |
| $R_{k,m}$ | Measured radiance in channel k |
| $T_{base}$ | Temperature at cloud base |
| $T_c$ | Centroid temperature, i.e. thermodynamic temperature at centroid altitude $Z_c$ |
| $T_{k,BB}$ | Blackbody brightness temperature in channel k |
| $T_{k,BG}$ | Background brightness temperature in channel k |
| $T_{k,m}$ | Measured brightness temperature in channel k |
| $T_r(k)$ | Radiative temperature in channel k |
| $T_{top}$ | Temperature at cloud top |
| $\tau_{a,k}$ | Effective absorption optical depth in channel k |
| $\tau_{vis}$ | Visible optical depth |
| $WF_{IIR}$ | IIR weighting function |
| $Z_c$ | Centroid altitude of the 532-nm attenuated backscatter |

Line 20: the authors discuss reducing biases found at very small emissivities in V3 of their products, both here and in Section 3.2.1 beginning at line 190. My interpretation of this is that there is a significant low biases in the ice cloud microphysical indices at very low values of the cloud emissivity, which is the same thing as stating that there is a bias at very low ice cloud optical depths.
Yes, we agree.

On lines 26/27, the authors state that V4 improved retrievals in ice clouds having large optical depths. My point is to be consistent in the use of cloud emissivity or cloud optical depth. In fact, lines 569-570 say this very clearly: "The IIR Level 2 algorithm has been modified in the V4 data release to improve the accuracy of the microphysical indices in clouds of very small (close to 0) and very large (close to 1) effective emissivities." Perhaps this sentence should also be in the Abstract.
We added this sentence in the abstract as suggested. The sentence is now inserted on line 20.

And on line 26, we replaced
"We have also aimed at improving retrievals in ice clouds having large optical depths by refining the determination of the radiative temperature needed for emissivity computation."

with (changes in italic)

"We have also *improved* retrievals in ice clouds having large *emissivity* by refining the determination of the radiative temperature needed for emissivity computation."

Line 25: why is the IIR channel at 8.65 microns written as 08.65 here and throughout the manuscript? Is there a reason for including a leading zero on this wavelength?
We wrote 08.65 µm for this wavelength because we chose the short notation '08' to designate this channel, which itself was chosen to have the same number of digits as in '10' and '12' channels. We recognize that we could have used a different approach.

Line 26: suggest changing "aimed at improving" to "improved"
Done

Line 31: define what is meant by "dense ice clouds" here and on lines 112, 122, 296, and 587.
On lines 31 and 587, the sentences were confusing, and we deleted "dense ice clouds".

On line 31, the new sentence reads:
"As shown in Part II, this improvement reduces the low biases at large optical depths that were seen in V3 and increases the number of retrievals."

And on line 587, the new sentence reads (changes in italic):
"This correction is expected to *both* increase the number of valid retrievals of crystal sizes *and reduce biases for ice clouds of* large optical depth*."*

On line 112, we are not using the term "dense" anymore and the sentence now starts as (changes in italic):
"The rationale is that unless these low layers are dust (or volcanic ash) layers *of sufficient optical depth*, …"

On line 122, we clarified by changing the sentence to (changes in italic):
"However, clouds detected at single shot resolution have large signal-to-noise ratios (SNR), indicating that *their optical depth is likely large and that they* actually should not be ignored."

Finally, on line 296, the end of the sentence now reads (changes in italic):
"…., and a marked negative tail down to about -8 K is observed, because *these cleared clouds have a fairly large optical depth* and are often colder than the surface".

Line 33: mostly a comment: this is the first of 24 references to "ice crystal models" or something similar in the text. The term "crystal" generally suggests a pristine shape such as a column or plate. The term "particle" includes all habits, pristine or very complex. Naturally occurring ice particles mostly defy description. This article more properly describes the adoption of two "ice habit models" composed of either single hexagonal columns (first found on line 33) or aggregates of columns (line 34).
Thank you.
We changed "ice crystal model" to "ice habit model".

Line 41: add a sentence to provide background and a reference for the A-Train for those readers who may not be familiar with it.
The sentence now reads (changes in italic):
"The A-Train *international constellation of satellites (Stephens et al., 2002)* has delivered a broad range….."

Line 41: define spectrum of wavelengths meant by visible and infrared
We now write:
"…operating in the *visible/near infrared (0.4 – 8 μm) and infrared (8-15 μm)*…."

Line 42: suggest changing "combination of infrared" to "combination of passive infrared"
Done

Lines 127-128: provide a description of the new types of scenes that have been introduced when at least one cleared cloud is present in the column
These new scene types are meant to identify the scenes that are cloud-free according to the 5-km layer products, but have at least one cleared cloud in the column. No IIR retrievals are attempted for these new scene types.
We tried to clarify the text, which now reads:

"Cloud-free scenes in V4 are pristine and have no single shot cleared clouds, while new types have been introduced *to identify scenes that are cloud-free according to the 5-km layer products, but have at least one cleared cloud in the column. No IIR retrievals are attempted for these new scene types*."

Lines 185 and 186: define what is meant specifically by optically very thin and very thick cloud here.
We revised the sentence, and these terms are not used anymore. It now reads (changes in italic):
"Because the sensitivity of the split-window technique decreases *as effective emissivity approaches 0 and 1*, $\Delta\varepsilon_{eff}12$-k is supposed to tend towards zero on average when $\varepsilon_{eff,12}$ tends towards 0 and towards 1."

Line 218: suggest changing "Earth Surface" to "surface"
Done

Line 218: interpolated atmospheric profiles: how many layers/levels are in the interpolated profiles? Do these profiles include trace gases? Are these augmented at very high altitudes by a climatology, e.g., for ozone if this is part of the profile?
We meant to say that the profiles are interpolated horizontally and temporally.
The atmospheric profiles are from the 72 levels of the MERRA-2 model. These profiles are temperature, specific humidity and ozone profiles.
The beginning of Sect. 3.3.1 now reads (changes in italic):

*"*The background radiance from the *surface* is computed using the FASRAD model fed by *horizontally and temporally* interpolated *temperature, water vapor, and ozone* profiles and skin temperatures. *These ancillary data are from the MERRA-2 reanalysis products in V4.*

Line 222: suggest changing "thanks to the advances" to "to take advantage of recent advances"
Done

Section 3.4.1, lines 313-317: the discussion on calculating a centroid altitude and temperature for multi-layered cloud cases is a bit confusing to me. If a vertical column contains optically thin ice cloud overlying a low-level water cloud, can the resulting centroid be in the mid-troposphere where there is no cloud layer? If this is a possibility, there should be a flag provided to indicate that multilayered clouds are present for that retrieval so that these cases can be filtered out if so desired. More specifically, the flag should be provided with the cloud properties such as the centroid altitude and temperature so that a user does not have to look at potentially multiple products (e.g., cloud mask or cloud phase) to find this detail. The availability of a flag would certainly be of help when comparing your cloud product to a simulated cloud field based on, for example, large eddy simulations.
Thank you for this question.
You are correct that if a vertical column contains optically thin ice cloud overlying a low-level semi-transparent water cloud, the resulting centroid can be in the mid-troposphere where there is no cloud layer.

A lot of CALIOP parameters describing the cloudy scenes are reported in the IIR product. For instance, we report an Ice_Water_Flag, which tells the user if the column includes only ice clouds or only water clouds, etc..., and we also report the CALIOP confidence in the feature type and phase assignments. A case with a thin ice clouds overlying a low-level semi-transparent cloud is flagged as mixed. There is also a flag (i.e., the Multi_Layer_Flag) specifying the number of layers selected by the IIR algorithm in the column. The full list of parameters reported in the IIR Level 2 products is available at:
https://www-calipso.larc.nasa.gov/resources/calipso_users_guide/data_summaries/iir/cal_iir_l2_track_v4-20_desc.php

We added the following text at the end of Sect. 2 about the scene classification:

*"A lot of other parameters characterizing the scenes are reported in the V4 IIR product. Among them are the number of layers in the cloud system, as well as an "Ice Water Flag" which informs the user about the phase of the cloud layers included in the system, as assigned by the V4 CALIOP Ice/Water phase algorithm (Avery et al., 2020). A companion "Quality Assessment" flag reports the mean confidence in the feature type (i.e., cloud or aerosol) classification (Liu et al., 2019) and in the phase assignment for these cloud layers. The product also includes the number of tropospheric dust layers and of stratospheric aerosols layers in the column and the mean confidence in the feature type classification. All the suitable scenes are processed regardless of the confidence in the classifications and phase assignments reported in the CALIOP products, so that the user can define customized filtering criteria adapted to specific research objectives."*

In Section 4 somewhere, it would be quite interesting to know the range of the effective diameter (De) values for ice clouds inferred from both V3 and V4. Does the range change between V3 and V4? Additionally, does the range ever approach the boundaries of the LUT, either very low or very high values? How often does this happen?

These questions are addressed in details in Sect. 3 of the companion "Part II" paper available at amt.copernicus.org/preprints/amt-2020-388/, preprint accepted for publication in the AMT journal.

As a result of the improved accuracy of $\beta_{eff}12/10$ and $\beta_{eff}12/08$, the consistency between $D_e12/10$ and $D_e12/08$ is drastically improved in V4 at $\varepsilon_{eff,12}$ smaller than 0.5 when the background radiance is computed using the radiative transfer model and cannot be derived from neighboring observations, which represents about 70 % of the cases.

In V4, both $\beta_{eff}12/10$ and $\beta_{eff}12/08$ are in the range of expected values (according to the respective LUTs) more than 80 % of the time for $\varepsilon_{eff,12}$ between 0.05 and 0.80. In contrast, the $\varepsilon_{eff,12}$ 80 % range in V3 was only 0.15 – 0.7 for the 12/10 pair and only 0.25 – 0.7 for the 12/08 pair.

Most of the time, failed retrievals are due $\beta_{eff}$ found smaller than the lower boundary of the LUT.

The $\beta_{eff}12/10$ and $\beta_{eff}12/08$ indices are typically larger in V4 than in V3, which decreases $D_e$ in V4 for a given LUT, but the V4 LUTs tend to provide a larger $D_e$ than in V3 for a given value of $\beta_{eff}$.

As a result, we find that $D_e12/10$ is not significantly changed in V4 compared to V3. However, $D_e12/08$ is smaller in V4 by up to 15 µm at $\varepsilon_{eff,12} < 0.2$, and larger by up to 10 µm at $\varepsilon_{eff,12}$ between 0.2 and 0.9. $D_e$ is therefore smaller in V4 by up to 7.5 µm at $\varepsilon_{eff,12} < 0.2$ and larger by up to 5 µm at $\varepsilon_{eff,12}$ between 0.2 and 0.9.